# Credit Assignment via Neural Manifold Noise Correlation

**Byungwoo Kang** [1 2]   **Maceo Richards** [1 2]   **Bernardo Sabatini** [1 2]

## Abstract

Credit assignment, the process of determining how changes in individual neurons and synapses influence a network's output, is central to learning in brains and machines. Noise correlation-based methods, which estimate gradients by correlating perturbations of activity with changes in output, provide a biologically plausible solution to credit assignment but scale poorly as accurately estimating the Jacobian requires that the number of perturbations scale with network size. Moreover, isotropic noise conflicts with neurobiological observations that neural activity lies on a low-dimensional manifold. To address these drawbacks, we propose *neural manifold noise correlation* (NMNC), which performs credit assignment using perturbations restricted to the neural manifold. We show theoretically and empirically that the Jacobian row space aligns with the neural manifold in trained networks, and that manifold dimensionality scales slowly with network size. NMNC substantially improves performance and sample efficiency over vanilla noise correlation in convolutional networks trained on CIFAR-10, ImageNet-scale models, and recurrent networks. NMNC also yields representations more similar to the primate visual system than vanilla noise correlation. These findings offer a mechanistic hypothesis for how biological circuits could support credit assignment, and suggest that biologically inspired constraints may enable, rather than limit, effective learning at scale.

## 1. Introduction

The problem of determining how individual neurons and synapses contribute to a network's output, often referred to as the credit assignment problem, is fundamental to learning in both artificial and biological neural networks (Minsky, 1961; Rumelhart et al., 1986). Backpropagation solves this elegantly but requires biologically implausible features: symmetric forward and backward weights, distinct forward and backward passes, and segregation of forward and backward pass activations (Crick, 1989; Grossberg, 1987). At its essence, credit assignment requires estimating the Jacobian of the network, the gradients of the network output with respect to hidden unit activations.

Noise correlation-based methods estimate gradients by injecting noise and correlating it with output changes (Williams, 1992; Werfel et al., 2003). Unlike feedback alignment approaches (Lillicrap et al., 2016; Nøkland, 2016), they directly approximate forward-pass gradients with local learning rules. The simplest form, weight perturbation, perturbs individual synaptic weights and observes changes in output (Jabri & Flower, 1992). A more efficient variant, node perturbation, perturbs neural activities rather than individual weights (Williams, 1992; Fiete & Seung, 2006). This approach underlies many modern attempts to develop biologically plausible learning rules (Bartunov et al., 2018; Kunin et al., 2020; Meulemans et al., 2021; 2022a). However, the number of perturbations required to accurately estimate the Jacobian scales with network size (Werfel et al., 2003; Ren et al., 2023). In addition, isotropic noise conflicts with neurobiological evidence that neural activity (not only task-related activity but also its trial-to-trial variability and spontaneous activity) lies on a low-dimensional manifold often referred to as *neural manifold* (Cunningham & Yu, 2014; Huang et al., 2019; Lin et al., 2015; Gardner et al., 2022; Chaudhuri et al., 2019; Dimakou et al., 2025; Luczak & MacLean, 2012; Luczak et al., 2009; Kenet et al., 2003; Engel & Steinmetz, 2019).

In this work, we connect these two observations and examine if *we can exploit the structure of the neural manifold to make noise correlation scalable.* We provide theoretical and empirical evidence that the gradients lie approximately within the same low-dimensional manifold as the activity itself. In addition, we show that manifold dimensionality

---

[1]Department of Neurobiology, Howard Hughes Medical Institute, Harvard Medical School, Boston, MA, USA [2]Kempner Institute for the Study of Natural and Artificial Intelligence, Harvard University, Boston, MA, USA. Correspondence to: Byungwoo Kang <byungwoo_kang@hms.harvard.edu>.

*Proceedings of the 43rd International Conference on Machine Learning*, Seoul, South Korea. PMLR 306, 2026. Copyright 2026 by the author(s).

scales slowly with network size. Based on these motivations, we propose *neural manifold noise correlation* (NMNC), which estimates a neural manifold online, injects noise along the manifold directions, and correlates output fluctuations with the low-dimensional noise. We evaluate NMNC on deep convolutional networks, ImageNet-scale models, and recurrent neural networks, demonstrating substantial improvements over vanilla noise correlation in performance and sample efficiency. Finally, we show that training convolutional networks with NMNC yields more primate-like visual representations than with vanilla noise correlation.

## 2. Background and motivation

### 2.1. Alignment of Jacobian row space and neural manifold

Consider the Jacobian $J_l = \frac{\partial \mathbf{y}}{\partial \mathbf{x}_l} \in \mathbb{R}^{n_o \times n_l}$, which maps perturbations in layer $l$'s activation space to changes in the network output $\mathbf{y}$. Let $\mathcal{M}_l \subset \mathbb{R}^{n_l}$ denote the neural manifold at layer $l$, and let $U_l \in \mathbb{R}^{n_l \times d_l}$ be an orthonormal basis for the subspace spanned by $\mathcal{M}_l$. Any perturbation decomposes into components parallel and orthogonal to the manifold:

$$\boldsymbol{\xi} = U_l U_l^T \boldsymbol{\xi} + (I - U_l U_l^T)\boldsymbol{\xi} = \boldsymbol{\xi}_\parallel + \boldsymbol{\xi}_\perp \qquad (1)$$

The central point is that the network's downstream layers have been trained exclusively on activations drawn from $\mathcal{M}_l$. Consequently, the network's response to $\boldsymbol{\xi}_\perp$ is essentially undefined and driven by the random structure at initialization. In contrast, $\boldsymbol{\xi}_\parallel$ probes the part of $J_l$ learned during training that captures the meaningful input-output relationships.

A complementary view comes from tracking how gradients shape downstream weights. Note that

$$J_l = \frac{\partial \mathbf{y}}{\partial \mathbf{x}_l} = \frac{\partial \mathbf{y}}{\partial \mathbf{s}_{l+1}}\frac{\partial \mathbf{s}_{l+1}}{\partial \mathbf{x}_l} = \frac{\partial \mathbf{y}}{\partial \mathbf{s}_{l+1}}W_{l+1}, \qquad (2)$$

where $\mathbf{s}_{l+1} = W_{l+1}\mathbf{x}_l$ and $\mathbf{x}_l = \phi(\mathbf{s}_l)$. Because gradient descent updates $W_{l+1}$ with $\Delta W_{l+1} \propto \boldsymbol{\delta}_{l+1}x_l^T$, the row space of the learned part of $W_{l+1}$ is spanned by the presynaptic activations $x_l^T$. Thus, via (2), it follows that the row space of $J_l$ is spanned by the history of the $x_l^T$, up to the random initial component.[1] Since training drives the rows of the Jacobian to preferentially align with the neural manifold (even if not fully contained within a strictly low-dimensional approximation of it), probing manifold directions should capture most of the gradient information relevant for learning.

We empirically confirmed the above theoretical considerations in a convolutional neural network trained by back-

propagation on CIFAR-10 (Figure 1) (see Section B.1 for architecture details).[2] With training, the Jacobian aligns with the neural manifold (defined by PCA), leading to significant fractions of its variance explained by a relatively small number of principal components (PCs).[3]

### 2.2. Scaling of neural manifold dimensionality with network size

If the neural manifold dimension $d_l$ remains small as networks scale, then restricting perturbations to this subspace can improve the sample efficiency of noise correlation. We therefore varied network width over two orders of magnitude (see Section E for details), holding architecture and dataset fixed, and estimated intrinsic dimensionality using TwoNN (Facco et al., 2017; Sharma & Kaplan, 2022) (see Section G for a self-contained explanation) and PCA. Across layers, manifold dimensionality grows slowly with width and remains far below $n_l$ (Figure 2), suggesting that the *effective* dimensionality of credit assignment can be much smaller than the raw activation dimension especially in large neural networks.

We also examined depth as an additional scaling axis. Depth scaling is less straightforward to interpret than width scaling: neural manifolds are naturally defined layer-wise, reflecting features at a specific level of abstraction, and scaling width directly probes representational capacity within a layer. On the other hand, scaling depth changes the number of nonlinear transformations, the effective receptive fields, and thus the representational role of each measured layer. Moreover, multiple valid depth-scaling schemes could exist, leading to interpretive ambiguity.

Nevertheless, to understand how the neural manifold dimensionality scales along this important additional axis, we performed a controlled depth-scaling experiment in which each

---

[1]A similar observation has previously been made in (Singhal et al., 2023).

[2]An alignment between the Jacobian and neural manifold was also observed in an AlexNet trained on ImageNet (Figure 8).

[3]We note that substantially more PCs are required to capture the variance of the Jacobian than that of the activations. One possible explanation is that, in Figure 1, PCA is performed on the current activations at each epoch, even though the learned part of the Jacobian is shaped by the history of activations during training. However, repeating the analysis with incremental PCA over activation history only modestly increased the fraction of Jacobian variance explained by the top 500 PCs at epoch 100, from 0.58/0.53/0.33/0.88 to 0.61/0.53/0.35/0.92 for Conv1/2/3/FC1. Thus, using current activations slightly understates the alignment between the Jacobian and neural manifold, but does not account for the larger number of PCs needed to capture Jacobian variance. The remaining difference likely arises because the Jacobian is also determined by downstream weights, which are in turn shaped by the history of downstream activations and error signals. This effectively changes the relative importance of different PCs and makes low-variance PCs account for a substantial fraction of the Jacobian variance, much more than they do for the activation variance.

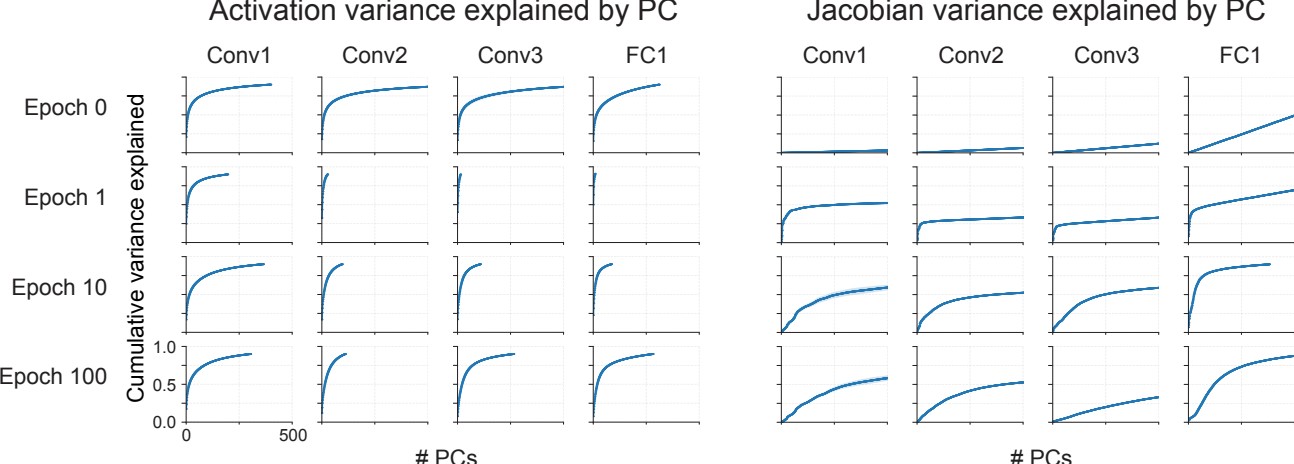

*Figure 1.* (Left) Variance of activations explained by principal components across training epochs for each layer of convolutional neural networks trained on CIFAR-10. Epoch 0 refers to the network prior to training, and Epoch 100 is the last epoch. (Right) Same analysis applied to the Jacobian variance. Curves are shown up to 90% cumulative variance explained. The number of hidden units are $16384, 8192, 4096, 1024$ for Conv1, Conv2, Conv3, and FC1 layers, respectively. See mean $\pm$ std, $n = 5$ seeds.

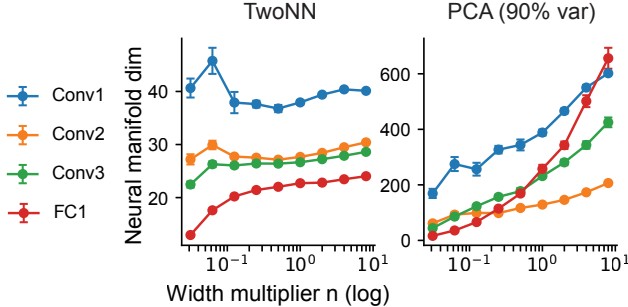

*Figure 2.* Network size vs. neural manifold dimensionality. (TwoNN or #PCs for 90% variance). The numbers of hidden units are $16384n, 8192n, 4096n, 1024n$ for Conv1, Conv2, Conv3, and FC1 layers, respectively. Mean $\pm$ std, $n = 5$ seeds.

convolutional stage was repeated $n$ times with extra layers using stride 1 (i.e. no additional downsampling) to preserve the size of activations at that stage.[4] Then, we measured the manifold dimensionality at the end of each convolutional stage and FC1. We found that the manifold dimensionality increases with depth in Conv1, but generally decreases in subsequent layers, consistent across PCA and TwoNN (Table 7). Importantly, test accuracy remained stable across settings ($\sim 70\%$), suggesting this is not due to optimization failure. Overall, these results indicate that, beyond early layers, increasing depth does not necessarily increase the effective dimensionality of downstream representations, and can instead promote dimensionality compression, complementing our width-scaling results.

---

[4]For example, for a 2x depth scaling, the architecture becomes Conv1 → Conv1 (stride 1) → Conv2 → Conv2 (stride 1) → Conv3 → Conv3 (stride 1) → FC1.

## 3. Neural manifold noise correlation

Based on the above theoretical and empirical motivations, we propose that credit assignment can be performed using *neural manifold noise correlation* (NMNC). NMNC learns feedback weights by performing noise correlation *within* each layer's activity manifold. For each layer $l$, we maintain (i) a low-dimensional basis $U_l \in \mathbb{R}^{n_l \times d_l}$ (estimated online from activations) and (ii) feedback weights $B_l \in \mathbb{R}^{n_l \times n_o}$.

**Feedback learning (noise correlation).** For every $b$ training iterations, we update $U_l$ (using incremental PCA (Ross et al., 2008)), sample $\zeta_l \sim \mathcal{N}(0, I_{d_l})$, form a manifold-restricted perturbation $\xi_l = U_l \zeta_l$, and run a noisy forward pass to obtain $\Delta \mathbf{y} = \tilde{\mathbf{y}} - \mathbf{y}$. We then update the feedback weights with an exponential moving average:

$$B_l \leftarrow (1 - \eta_B) B_l + \eta_B \cdot \frac{1}{N_b} \left( \xi_l \Delta \mathbf{y}^T \right), \qquad (3)$$

where $N_b$ is batch size. Full pseudocode is given in Algorithm 1 (Section A).

**Forward-weight updates.** Given the current $B_l$, we compute a pseudo-error at each layer from the output error $\delta_{\text{out}}$:

$$\delta_l = \phi'(\mathbf{s}_l) \odot (B_l \delta_{\text{out}}), \qquad (4)$$

and update forward weights locally via $\Delta W_l = -\eta \, \delta_l \mathbf{x}_{l-1}^T$.

**Key differences from vanilla noise correlation.** Vanilla noise correlation (VNC) samples isotropic noise $\xi_l \sim \mathcal{N}(0, \sigma^2 I_{n_l})$ in the full activation space. NMNC samples noise in the low-dimensional manifold ($d_l$ dimensions) and

then projects it to the full space via $U_l$. This reduces variance and improves sample efficiency. To ensure fair comparison, we match noise magnitudes in NMNC and VNC: $\sigma_{\text{VNC}} = \sqrt{d_l/n_l}\,\sigma_{\text{NMNC}}$.

## 4. Experiments

We evaluate NMNC as a scalable, perturbation-based credit assignment mechanism across three regimes: (i) direct-feedback learning in a convolutional network trained on CIFAR-10, (ii) ImageNet-scale training using AlexNet with layerwise feedback (Weight Mirror), and (iii) recurrent networks trained via weight perturbation. Unless otherwise stated, we use a shared forward-weight optimizer and training recipe across methods within each benchmark to enable controlled comparisons of credit-assignment rules. The output layer is always trained using exact gradients (see Section B for architectures and Section C for training details).

### 4.1. Performance and sample efficiency of NMNC

**Setup.**    We train the CIFAR-10 convolutional network described in Section B.1 using (i) standard backpropagation, (ii) direct feedback alignment (DFA; fixed random feedback), (iii) an "InitJac" baseline with fixed feedback weights set to the *initial* Jacobian (evaluated on random Gaussian inputs), and (iv) learned-feedback variants using VNC or NMNC. For NMNC, the neural manifold basis at each hidden layer is estimated online via incremental PCA, and perturbations are restricted to that subspace. For both NMNC and VNC, we considered "InitJac" and "No InitJac" variants: in the former, feedback weights are initialized to the initial Jacobian, whereas in the latter they are initialized randomly. In both cases, feedback weights are learned during training.

**Main comparison.**    Figure 3A shows that NMNC substantially outperforms VNC and DFA and approaches backpropagation performance. This improvement is obtained under a fixed perturbation budget: both VNC and NMNC learn feedback weights via noise correlation, but NMNC concentrates perturbations along directions that are most functionally relevant for the trained network (as motivated in Figure 1). NMNC also outperformed random subspace controls in which noise is injected into a fixed or resampled (every time we inject noise) random subspace of the same dimensionality as the neural manifold used by NMNC (Figure 9), suggesting that the improvement is not merely due to the reduced dimensionality of perturbations. We emphasize that our comparisons focus on learning efficiency rather than asymptotic performance. While VNC can continue to improve with additional perturbation samples, NMNC reaches higher accuracy within the same training horizon and update schedule.

**InitJac vs. No InitJac**    Although InitJac variants show early advantages, learned-feedback methods (NMNC/VNC) and DFA close this gap as training progresses. We therefore default to the No-InitJac setting for the remainder of the study, isolating the effect of how perturbations are sampled (full-space versus manifold-restricted).

**Sample efficiency vs. feedback-update frequency.**    Increasing the update interval $b$ reduces the number of perturbation samples used to learn the feedback weights. Figure 3B and C show that NMNC maintains higher accuracy than VNC across a wide range of $b$, consistent with reducing the effective dimensionality of the estimation problem from $n_l$ to $d_l \ll n_l$.

### 4.2. Mechanisms underlying NMNC's advantage over VNC

To better understand why NMNC improves learning, we compare the *pseudo-gradients* induced by learned feedback weights to the true backpropagation gradients (computed solely for this analysis and not used for training). Across layers, we find two consistent effects:

**Mechanism 1: improved early gradient alignment.**    Figure 4A and C show that NMNC yields better alignment to the true gradient (smaller angle) early in training and in lower layers, which have higher-dimensional activity space. In some settings, VNC can match or slightly exceed NMNC later in training when feedback updates are frequent and enough samples accumulate. However, probably because alignment is more important earlier in learning, NMNC's early alignment advantage translates into better performance, even when VNC catches up later in terms of alignment.

**Mechanism 2: larger effective step along the true gradient.**    We also quantify the magnitude of the pseudo-gradient component that lies along the true gradient direction. Figure 4B shows that NMNC produces a larger projected pseudo-gradient across layers, meaning that NMNC typically takes a larger effective step in the direction that decreases the loss. Interestingly, we observe regimes in which VNC attains higher cosine alignment late in training even though NMNC still has a larger projected magnitude; we provide an explanation below.[5]

_______________

[5]Given that NMNC can produce pseudo-gradients with larger norm than VNC, one might ask whether VNC could compensate simply by increasing the forward learning rate. Empirically, we found that larger learning rates for VNC often destabilize training, consistent with additional stochasticity from SGD and sample-to-sample variability in gradient estimation.

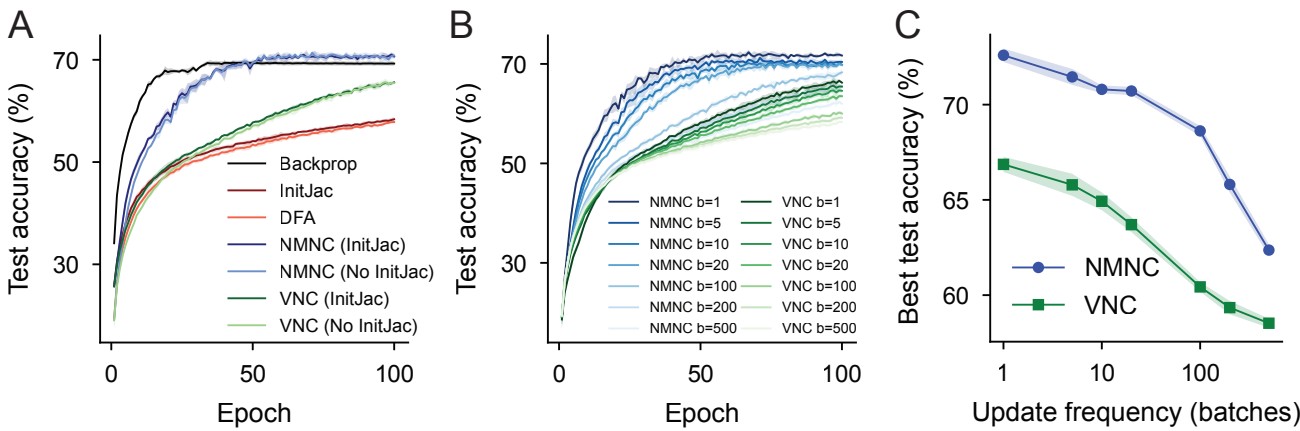

*Figure 3.* Performance and sample efficiency of NMNC and VNC on CIFAR-10. (A) Test accuracy vs. epochs for different learning rules. (B) Test accuracy vs. epochs for varying frequencies of feedback update for NMNC and VNC (No InitJac). Feedback weights are updated every $b$ batches. (C) Best test accuracy vs. noise correlation frequency. Same data as (B). Mean $\pm$ std, $n = 5$ seeds.

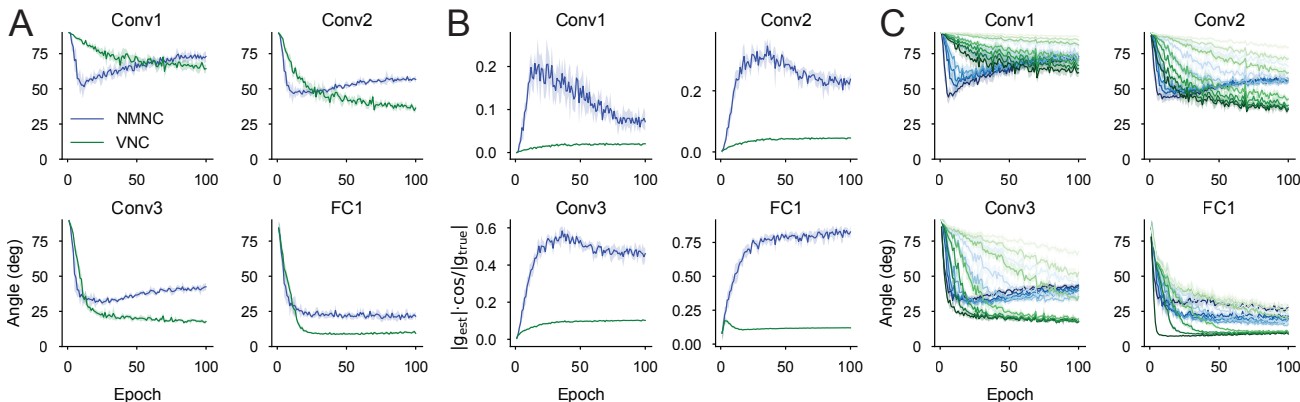

*Figure 4.* Alignment between true and estimated gradients in activation space (see Figure 10 for alignment in weight space). (A) Angle between the true and estimated gradients for NMNC and VNC across layers. (B) Normalized magnitude of the estimated gradient projected onto the true gradient direction for NMNC and VNC across layers. (C) Same as (A) but for varying frequencies of feedback update. The color scheme is the same as in Figure 3B. mean $\pm$ std, $n = 5$ seeds.

**Relationship between pseudo- and true gradient.** For small perturbations $\Delta y \approx J_l \xi$, the expected feedback-weight update in Algorithm 1 is

$$\mathbb{E}[\Delta B_l] = \mathbb{E}[\xi \Delta y^\top] \approx \mathbb{E}[\xi \xi^\top] J_l^\top = \Sigma_l J_l^\top, \quad (5)$$

where $\Sigma_l := \mathbb{E}[\xi \xi^\top]$ is the noise covariance. Ignoring the slow drift of $J_l$, the feedback weights converge to $B_l^\star \propto \Sigma_l J_l^\top$ (see Algorithm 1), and the pseudo-gradient is

$$\tilde{g}_l = B_l^\star \delta_{\text{out}} \approx \Sigma_l J_l^\top \delta_{\text{out}} = \Sigma_l g_l. \quad (6)$$

Thus noise correlation returns the true gradient pre-multiplied by the noise covariance.

**An explanation for early alignment.** Let $g_l$ denote the true backprop gradient with respect to the activations of layer $l$, and let $\xi_l$ be the noise injected into that layer when learning the feedback weights. After $k$ noise-correlation

updates, the resulting pseudo-gradient can be written as

$$\tilde{g}_l^{(k)} = \hat{\Sigma}_l^{(k)} g_l, \qquad \hat{\Sigma}_l^{(k)} = \frac{1}{k} \sum_{i=1}^{k} \xi_l^{(i)} \xi_l^{(i)\top}, \quad (7)$$

where $\hat{\Sigma}_l^{(k)}$ is the empirical noise covariance. Its expectation is the true covariance $\Sigma_l = \mathbb{E}[\xi_l \xi_l^\top]$, so we may decompose

$$\tilde{g}_l^{(k)} = \Sigma_l g_l + \eta_l^{(k)}, \qquad \eta_l^{(k)} := (\hat{\Sigma}_l^{(k)} - \Sigma_l) g_l. \quad (8)$$

For Gaussian noise one can show (using standard fourth-moment identities) that

$$\mathbb{E}\big[\|\eta_l^{(k)}\|^2\big] = \frac{1}{k}\Big[\text{tr}(\Sigma_l)\, g_l^\top \Sigma_l g_l + g_l^\top \Sigma_l^2 g_l\Big]. \quad (9)$$

In VNC the noise is isotropic in the full $n_l$-dimensional activation space, $\Sigma_l^{\text{VNC}} = \frac{\tau_l}{n_l} I_{n_l}$, whereas in NMNC it

is restricted to the $d_l$-dimensional neural manifold with projector $P_l := U_l U_l^\top$, $\Sigma_l^{\text{NMNC}} = \frac{\tau_l}{d_l} P_l$, with $\tau_l = \mathbb{E}\|\xi_l\|^2$ matched between methods. Writing

$$\alpha_l := \frac{\|P_l g_l\|^2}{\|g_l\|^2} \in [0, 1] \tag{10}$$

for the fraction of gradient energy lying in the manifold, and using (9), we obtain

$$\mathbb{E}\|\eta_l^{(k)}\|_{\text{VNC}}^2 \approx \frac{\tau_l^2}{k n_l}, \qquad \mathbb{E}\|\eta_l^{(k)}\|_{\text{NMNC}}^2 \approx \frac{\tau_l^2 \alpha_l}{k d_l}. \tag{11}$$

Approximating $\eta_l^{(k)}$ as noise uncorrelated with signal $\Sigma_l g_l$ and replacing the norm of $\eta_l^{(k)}$ in the denominator by its expectation yields the following expressions for the expected squared cosine between the pseudo-gradient and the true gradient:

$$\mathbb{E}\left[\cos^2\left(\tilde{g}_l^{(k)}, g_l\right)\right]_{\text{VNC}} \approx \frac{k}{k + n_l + 1},$$
$$\mathbb{E}\left[\cos^2\left(\tilde{g}_l^{(k)}, g_l\right)\right]_{\text{NMNC}} \approx \frac{\alpha_l k}{k + d_l + 1}. \tag{12}$$

For small $k$, these scale as $\mathbb{E}[\cos^2]_{\text{VNC}} \approx k/n_l$ and $\mathbb{E}[\cos^2]_{\text{NMNC}} \approx \alpha_l k/d_l$, so NMNC has better early alignment whenever $\alpha_l > d_l/n_l$. In our CIFAR-10 setting, $d_l/n_l$ is small while the Jacobian row space is strongly aligned with the activity manifold (Figure 1), making this condition easy to satisfy. Intuitively, NMNC only needs to estimate a $d_l$-dimensional preconditioner that captures most of the gradient energy, whereas VNC must estimate an $n_l$-dimensional object from the same number of samples.

As $k \to \infty$, (12) predicts $\cos^2_{\text{VNC}} \to 1$ while $\cos^2_{\text{NMNC}} \to \alpha_l$, so VNC can eventually achieve slightly higher cosine alignment than NMNC if it accumulates many perturbation samples. In practice, with feedback updates every $b$ batches, each layer only sees $k \approx T/b$ samples over $T$ training iterations. When $T/b \ll n_l$ (e.g. larger $b$), VNC never reaches its asymptotic regime and NMNC maintains higher alignment throughout training, as observed in Figure 4C.

**An explanation for the pseudo-gradient magnitude and projection.** Noise correlation learns feedback weights proportional to $J_l \Sigma_l$, and therefore returns a pseudo-gradient proportional to $\Sigma_l g_l$. Under matched perturbation energy $\tau_l = \text{tr}(\Sigma_l)$, VNC yields $\tilde{g}_l^{\text{VNC}} = \frac{\tau_l}{n_l} g_l$ whereas NMNC yields $\tilde{g}_l^{\text{NMNC}} = \frac{\tau_l}{d_l} P_l g_l$. Their squared norms satisfy

$$\frac{\|\tilde{g}_l^{\text{NMNC}}\|}{\|\tilde{g}_l^{\text{VNC}}\|} = \frac{n_l}{d_l} \frac{\|P_l g_l\|}{\|g_l\|}. \tag{13}$$

Thus, whenever $\|P_l g_l\|/\|g_l\| > d_l/n_l$ (the same condition as above), NMNC produces a pseudo-gradient with larger

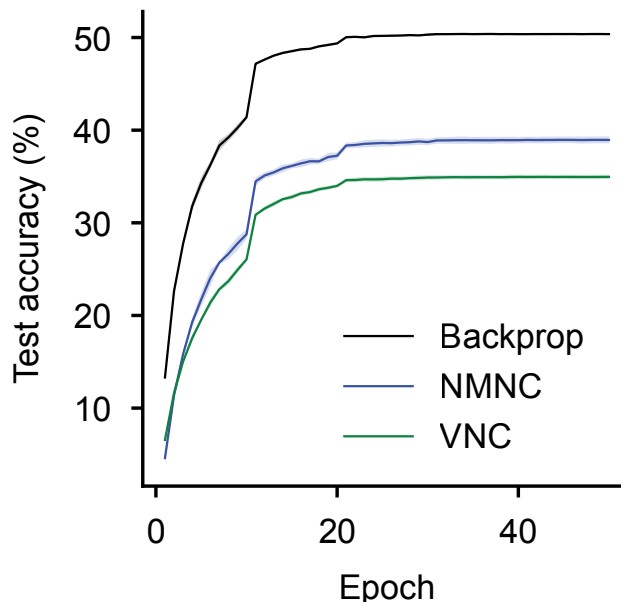

*Figure 5.* Comparison of NMNC and VNC on ImageNet. Test accuracy of AlexNet on ImageNet when trained with (i) Backprop, (ii) the weight mirror algorithm using vanilla noise correlation (VNC), and (iii) the weight mirror algorithm using neural manifold noise correlation (NMNC). mean $\pm$ std, $n = 5$ seeds.

expected norm. Moreover, the component along the true gradient direction scales as

$$\frac{\left\|\text{Proj}_{g_l}(\tilde{g}_l^{\text{NMNC}})\right\|}{\left\|\text{Proj}_{g_l}(\tilde{g}_l^{\text{VNC}})\right\|} = \frac{n_l}{d_l} \frac{\|P_l g_l\|^2}{\|g_l\|^2},$$

so NMNC can take a larger effective step along the true gradient even when its cosine alignment is slightly lower. This provides a parsimonious explanation for why, late in training, VNC can sometimes show higher alignment while NMNC continues to exhibit a larger projected pseudo-gradient (Figure 4B) and better learning.

### 4.3. Application of NMNC to ImageNet-scale models

Having established NMNC as a practical learning rule on CIFAR-10, we next test whether it can be applied at ImageNet scale. This regime is also relevant from a neuroscience perspective: training on large and diverse natural image datasets is associated with the emergence of more primate-like visual representations in deep networks (Conwell et al., 2024).

**Why AlexNet and why layerwise feedback.** We use AlexNet (rather than more recent architectures with very large activation tensors) because online incremental PCA becomes prohibitively expensive in our current implementation. Direct feedback from the output layer (as in the CIFAR-10 experiments) was unstable or substantially degraded, suggesting that a single linear map from output

error to early-layer activations is a poor approximation at this scale. We therefore adopt a *layerwise* feedback scheme based on the Weight Mirror approach (Akrout et al., 2019), and compare isotropic perturbations (VNC) to manifold-restricted perturbations (NMNC) (see Sections B.2 and C.2 for details).

**Results.** Training with NMNC significantly outperforms VNC on ImageNet, although a gap remains compared to backpropagation (Figure 5). This indicates that restricting perturbations to the neural manifold remains beneficial at this scale. We also observe that in this layerwise-feedback regime, VNC can exhibit higher cosine alignment than NMNC (Figure 12A). This may be because: 1. the feedback weights being learned in the layerwise regime are transposed convolutional kernels rather than direct output-to-hidden matrices, reducing the effective dimensionality of the estimation problem. 2. the feedback weights are updated every batch rather than every $b$ batches as in the CIFAR-10 experiment to achieve reasonable performance, giving VNC a large number of samples and thus pushing it closer to its asymptotic regime $\cos^2_{VNC} \to 1$ (as predicted by (12)). Despite this, NMNC yields larger pseudo-gradient magnitudes and projected steps, consistent with its improved accuracy (Figure 12B).

## 4.4. Neural representations of ImageNet-scale models trained with Backprop, NMNC and VNC

Beyond task performance, we examined whether restricting perturbations to the neural manifold influences the *representations* that emerge during learning. We analyzed the ImageNet-scale models trained with backpropagation, NMNC, and VNC.

**First-layer kernels.** A classic qualitative signature of ImageNet-trained AlexNet is that its first-layer convolutional kernels resemble Gabor filters, reminiscent of V1 receptive fields (Krizhevsky et al., 2012). Figure 6A shows that all three learning rules indeed broadly produce Gabor-like kernels. However, VNC-trained models additionally exhibit prominent salt-and-pepper, high-frequency patterns superimposed on the Gabor-like spatial profiles. This is reflected in the Fourier-domain visualization in Figure 6A (bottom row), which shows that VNC yields kernels with stronger high-frequency components. To address the potential confound due to difference in performance of the NMNC- and VNC-trained models, we also analyzed NMNC-trained models that are performance matched to the VNC-trained models (checkpoints at epoch 11). Their kernels still exhibited smooth Gabor-like spatial profiles with similar discrete Fourier transforms to their fully trained counterparts, and contained much less high-frequency components than the VNC-trained models (Table 6).

**Brain-score evaluation.** To more systematically compare representations to the primate ventral visual stream, we evaluated the trained models using Brain Score metrics for V4, IT, and behavior (Schrimpf et al., 2018; 2020). Figure 6B shows that backpropagation yields the highest Brain Scores overall, followed by NMNC and then VNC. As in the above analysis, the NMNC-trained models maintained higher Brain Scores than the VNC-trained counterparts, even when performance-matched. A higher Brain Score does not by itself imply that a learning rule is biologically correct, but the consistent ordering and the qualitative filter differences suggest that incorporating structure in perturbations (i.e. aligning them with natural activity patterns) can bias learning toward more brain-like representations.

## 4.5. Application of NMNC to recurrent neural networks

Biological circuits are highly recurrent, and credit assignment in recurrent neural networks (RNNs) poses an additional challenge: backpropagation through time (BPTT) requires transporting error signals *back in time* to the same neurons rather than to upstream layers. A naive application of node-perturbation-style noise correlation to RNNs would require learning feedback pathways that deliver appropriate temporal credit.

**Manifold-structured low-rank weight perturbation.** In the RNN setting, weight perturbation (WP) has a unique advantage: it does not require an explicit temporal feedback pathway, because it directly perturbs the recurrent weights and correlates the resulting loss change with the perturbation. Motivated by recent work on low-rank perturbation schemes for large-scale optimization (Sarkar et al., 2025), we tested whether *restricting* the perturbations to the neural manifold of the hidden state can improve WP in RNNs.[6]

Specifically, we compared: (i) standard full-rank WP with i.i.d. perturbations, (ii) rank-1 WP with i.i.d. factors, (iii) rank-1 WP whose factors lie in a fixed random subspace matched in dimension to the neural manifold, and (iv) rank-1 WP whose factors are sampled from the *neural manifold* of the hidden state (estimated online). For fair comparison, all perturbation types were scaled to have matched magnitude (see Section C.3 for details of architecture and training).

**Results.** On a sequential memory task, rank-1 manifold WP achieves the best performance among WP variants (Figure 7A). Consistent with the feedforward results, manifold-structured perturbations also yield better gradient estimates:

---

[6]Low-rank WP does not avoid the fundamental variance scaling of standard WP with the number of parameters (See Section F for an explanation); its main advantage is computational/hardware efficiency (Sarkar et al., 2025). Here we isolate a complementary effect: choosing the perturbation subspace to match the network's activity manifold.

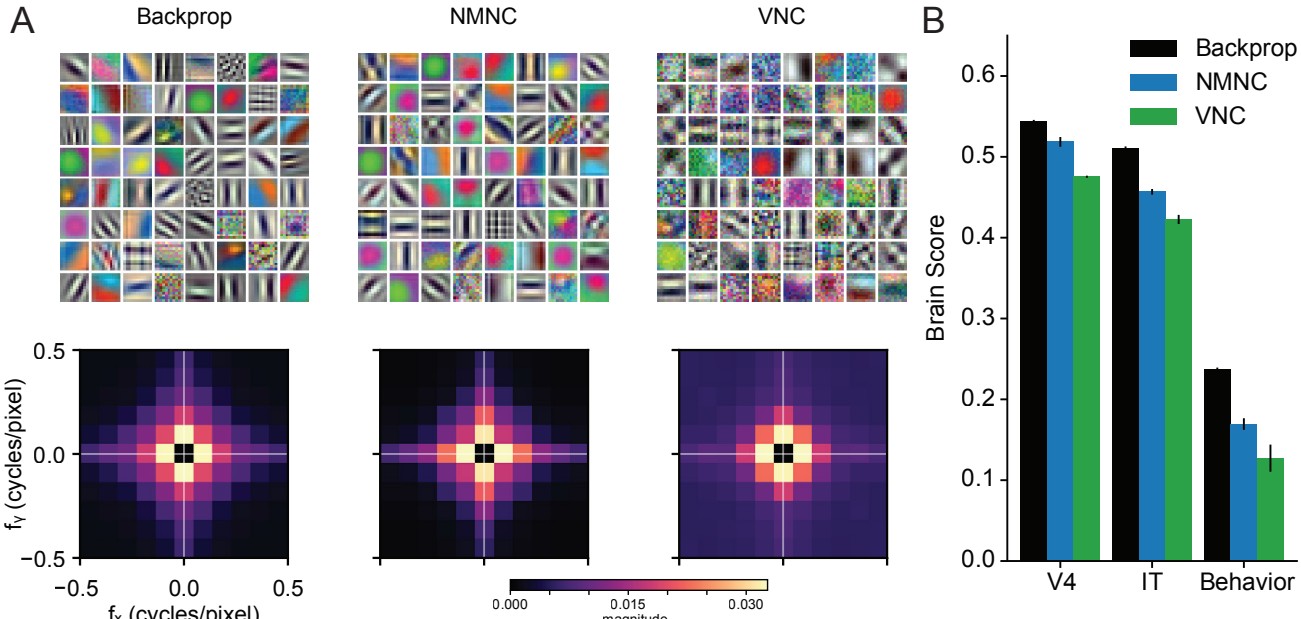

*Figure 6.* Comparison of neural representations emerging from NMNC and VNC. (A) (Top row) Conv1 kernels from models trained with Backprop, NMNC and VNC. (Bottom row) Discrete Fourier transform on the Conv1 kernels. mean, n = 5 seeds. (B) V4, IT, and Behavior Brain Scores for models trained with Backprop, NMNC and VNC. mean ± std, n = 5 seeds.

rank-1 manifold WP exhibits higher gradient alignment (Figure 7B) and larger projected pseudo-gradient magnitude (Figure 7C) for the recurrent weight matrix $W_{hh}$. These results suggest that the core NMNC principle of constraining perturbations to the neural manifold is applicable beyond feedforward networks and can improve perturbation-based learning in recurrent settings as well.

## 5. Discussion

Noise correlation-based methods offer an appealing route to biologically plausible credit assignment because they can estimate gradients using only forward computations and locally available information. However, their classic formulations using isotropic noise scale poorly as the number of samples required to obtain useful feedback signals scales with the dimensionality of the activity space.

**Neural manifolds reduce the effective dimensionality of credit assignment** Our central observation is that trained networks (and biological circuits) tend to operate on low-dimensional activity manifolds, and that the functionally relevant components of the Jacobian become aligned with these manifolds through learning (Figure 1). NMNC exploits this structure by performing noise correlation *on the manifold* rather than in the full activity space. This reduces the effective dimension of the estimation problem and improves sample efficiency. Empirically, NMNC closes much of the gap between perturbation-based learning and back-

propagation on CIFAR-10 (Figure 3), provides a consistent advantage over vanilla noise correlation at ImageNet scale when combined with layerwise feedback learning (Figure 5), and extends to recurrent networks via manifold-structured low-rank weight perturbation (Figure 7).

**Implications for neuroscience and biologically plausible learning.** From a biological standpoint, NMNC suggests a concrete hypothesis: correlated variability aligned with a circuit's intrinsic activity manifold may not be merely noise, but may provide the structured perturbations needed for credit assignment when combined with global output or performance signals (e.g. neuromodulators). Under this hypothesis, the brain may not need to inject independent perturbations across all neurons. Instead, it can exploit the low-dimensional structure of population activity to learn effective credit assignment with far fewer degrees of freedom.

**Limitations and future directions.** First, our current implementation uses PCA, which provides only a linear subspace approximation to potentially nonlinear manifolds. Incorporating nonlinear manifold models (e.g. learned encoders or locally linear subspaces) could further reduce bias while preserving sample efficiency. Second, online manifold estimation can be computationally expensive (or even infeasible) for modern architectures with very large activations; developing more efficient and hardware-friendly estimators would improve practical scalability. Third, the brain likely generates manifold-aligned fluctuations via mechanisms dif-

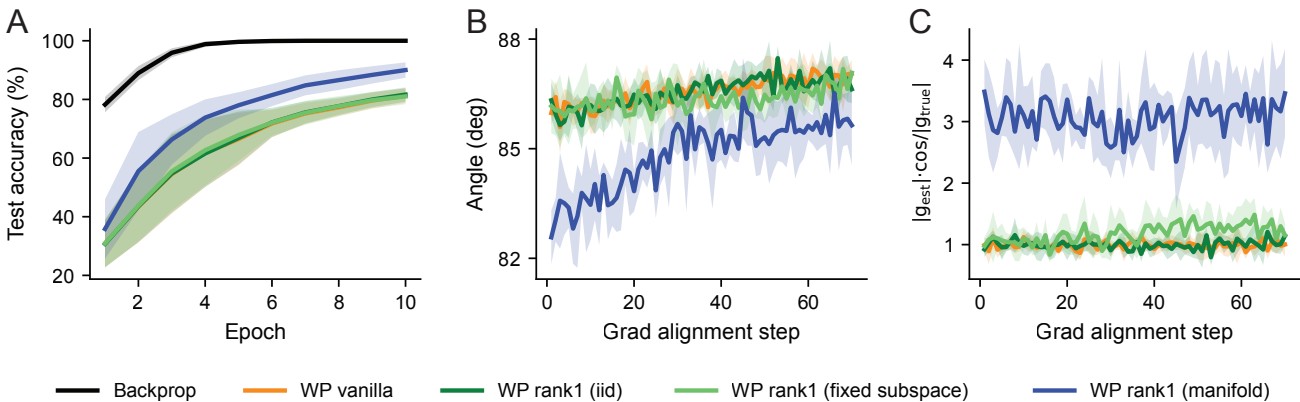

*Figure 7.* Neural manifold noise correlation in recurrent networks. (A) Performance of RNNs trained with Backprop and variants of weight perturbation (WP) on a sequential memory task. "WP vanilla" refers to standard full-rank WP with i.i.d. perturbations. "WP rank1 (iid)" uses rank-1 perturbations with i.i.d. factors. "WP rank1 (fixed subspace)" samples rank-1 perturbations from a fixed random subspace with dimensionality matched to the neural manifold. "WP rank1 (manifold)" samples rank-1 perturbations from the neural manifold of the hidden state. (B) Angle between the true and estimated gradient for $W_{hh}$. (C) Normalized magnitude of the estimated gradient projected onto the true gradient direction for $W_{hh}$. mean $\pm$ std, $n = 5$ seeds.

ferent from our specific implementation involving incremental PCA. Encouragingly, there are biologically plausible proposals for online PCA and related dimensionality reduction algorithms (Qiu et al., 2012; Oja, 1982; 1989; 1992; Kung & Diamantaras, 1990; Sanger, 1989; Foldiak, 1989; Linsker, 2005; Minden et al., 2018; Pehlevan et al., 2015). Incorporating such mechanisms into NMNC would make the algorithm more biologically realistic. Fourth, the remaining gap in performance on ImageNet and Brain Score indicates that complementary improvements in feedback parameterizations, initialization, and additional local learning signals are needed to close the gap with backpropagation. Fifth, while noise correlation-based methods including NMNC address the problem of symmetric forward and backward weights, one of the key biological implausibilities of backpropagation, they do not by themselves resolve the other problems mentioned in the introduction. Notably, several recent proposals that solve these additional issues also rely on noise correlation to learn feedback pathways, and we expect NMNC to similarly improve their sample efficiency relative to VNC (Meulemans et al., 2021; 2022a;b).

More broadly, our results indicate that incorporating biological constraints, rather than being a hindrance, can simplify the credit assignment problem and enable effective learning at scale. We hope NMNC motivates further work connecting low-dimensional population dynamics, structured variability, and plausible credit assignment mechanisms in both artificial and biological neural systems.

## Acknowledgements

We thank Gyuryang Heo, Rich Hakim, and Cengiz Pehlevan for reading and providing feedback on early drafts of the manuscript. This work was supported by NINDS R35NS137336. B.K. was supported by the Helen Hay Whitney Foundation, and M.R. was supported by Kempner baccalaureate fellowship. This work has been made possible in part by a gift from the Chan Zuckerberg Initiative Foundation to establish the Kempner Institute for the Study of Natural and Artificial Intelligence at Harvard University.

## Impact Statement

This work aims to advance understanding of how biologically plausible learning rules could support credit assignment in high-dimensional neural systems. By connecting perturbation-based learning to low-dimensional neural activity manifolds, it offers a conceptual bridge between empirical observations in neuroscience and learning algorithms studied in machine learning. The expected impact is primarily scientific, providing a framework and hypotheses that may guide future theoretical, computational, and experimental work at the interface of these fields. We do not anticipate direct societal risks specific to this contribution beyond those generally associated with progress in machine learning research.

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

# A. NMNC pseudocode

---

**Algorithm 1** Neural Manifold Noise Correlation (NMNC)

---

1: **Input:** Network with layers $1, \ldots, L$; manifold dimensions $\{d_l\}$; PCA update interval $b$; feedback learning rate $\eta_B$; batch size $N_b$
2: Initialize $\{U_l\}$ and $\{B_l\}$ randomly
3: **for** each training iteration $t$ **do**
4:     Forward pass: compute activations $\{\mathbf{x}_l\}$ and output $\mathbf{y}$
5:     *// Learn feedback weights via noise correlation:*
6:     **if** $t \mod b = 0$ **then**
7:         Update $\{U_l\}$ via incremental PCA on $\{\mathbf{x}_l\}$
8:         **for** each layer $l$ **do**
9:             Sample low-dim noise: $\boldsymbol{\zeta}_l \sim \mathcal{N}(0, I_{d_l})$
10:        Project to activation space: $\boldsymbol{\xi}_l = U_l \boldsymbol{\zeta}_l$
11:         **end for**
12:        Forward pass with noise: inject noise $\boldsymbol{\xi}_l$ to each layer and compute noisy output $\tilde{\mathbf{y}}$
13:        Compute output change: $\Delta \mathbf{y} = \tilde{\mathbf{y}} - \mathbf{y}$
14:        **for** each layer $l$ **do**
15:            Update: $B_l \leftarrow (1 - \eta_B)B_l + \eta_B \cdot \frac{1}{N_b}(\boldsymbol{\xi}_l \Delta \mathbf{y}^T)$
16:         **end for**
17:     **end if**
18:     *// Compute weight updates using learned feedback:*
19:     Compute output error: $\boldsymbol{\delta}_{\text{out}} = \frac{1}{N_b}(\text{softmax}(\mathbf{y}) - \mathbf{y}_{\text{target}})$
20:     **for** each layer $l = L, \ldots, 1$ **do**
21:         Compute pseudo-error: $\boldsymbol{\delta}_l = \phi'(\mathbf{s}_l) \odot (B_l \boldsymbol{\delta}_{\text{out}})$
22:         Update weights: $\Delta W_l = -\eta \cdot \boldsymbol{\delta}_l \mathbf{x}_{l-1}^T$
23:     **end for**
24: **end for**

---

# B. Network Architectures

## B.1. CIFAR-10 Architecture

For CIFAR-10 experiments, we use the same convolutional network architecture used in (Bartunov et al., 2018) (Table 1):

*Table 1.* CIFAR-10 network architecture. All convolutional and fully-connected layers (except the output layer) are followed by ReLU activations.

| Layer | Type | Input → Output | Kernel | Stride | Padding |
|---|---|---|---|---|---|
| conv1 | Conv2d | $3 \times 32 \times 32 \to 64 \times 16 \times 16$ | $5 \times 5$ | 2 | 2 |
| conv2 | Conv2d | $64 \times 16 \times 16 \to 128 \times 8 \times 8$ | $5 \times 5$ | 2 | 2 |
| conv3 | Conv2d | $128 \times 8 \times 8 \to 256 \times 4 \times 4$ | $3 \times 3$ | 2 | 1 |
| fc1 | Linear | $4096 \to 1024$ | – | – | – |
| fc2 | Linear | $1024 \to 10$ | – | – | – |

The activation shapes and corresponding flat dimensions for each layer are described in Table 2:

*Table 2.* Activation dimensions for each layer in the CIFAR-10 network.

| Layer | Activation Shape | Flat Dimension $n_l$ | Default # PCs $d_l$ |
|-------|------------------|----------------------|---------------------|
| conv1 | $(64, 16, 16)$ | 16,384 | 512 |
| conv2 | $(128, 8, 8)$ | 8,192 | 512 |
| conv3 | $(256, 4, 4)$ | 4,096 | 512 |
| fc1 | $(1024, )$ | 1,024 | 128 |

## B.2. ImageNet Architecture

For ImageNet experiments, we use a standard AlexNet architecture (Table 3):

*Table 3.* ImageNet (AlexNet) network architecture. All convolutional and fully-connected layers (except the output layer) are followed by ReLU activations.

| Layer | Type | Input $\rightarrow$ Output | Kernel | Stride | Padding |
|-------|------|----------------------------|--------|--------|---------|
| conv1 | Conv2d | $3 \rightarrow 64$ | $11 \times 11$ | 4 | 2 |
| pool1 | MaxPool2d | – | $3 \times 3$ | 2 | 0 |
| conv2 | Conv2d | $64 \rightarrow 192$ | $5 \times 5$ | 1 | 2 |
| pool2 | MaxPool2d | – | $3 \times 3$ | 2 | 0 |
| conv3 | Conv2d | $192 \rightarrow 384$ | $3 \times 3$ | 1 | 1 |
| conv4 | Conv2d | $384 \rightarrow 256$ | $3 \times 3$ | 1 | 1 |
| conv5 | Conv2d | $256 \rightarrow 256$ | $3 \times 3$ | 1 | 1 |
| pool5 | MaxPool2d | – | $3 \times 3$ | 2 | 0 |
| avgpool | AdaptiveAvgPool2d | Output: $6 \times 6$ | – | – | – |
| fc1 | Linear | $9216 \rightarrow 4096$ | – | – | – |
| fc2 | Linear | $4096 \rightarrow 4096$ | – | – | – |
| fc3 | Linear | $4096 \rightarrow 1000$ | – | – | – |

The activation shapes (before pooling where applicable) and corresponding flat dimensions are described in Table 4:

*Table 4.* Activation dimensions for each layer in the ImageNet network. Shapes reported are prior to any subsequent pooling operation.

| Layer | Activation Shape | Flat Dimension $n_l$ | Default # PCs $d_l$ |
|-------|------------------|----------------------|---------------------|
| conv1 | $(64, 55, 55)$ | 193,600 | 2,048 |
| conv2 | $(192, 27, 27)$ | 139,968 | 2,048 |
| conv3 | $(384, 13, 13)$ | 64,896 | 2,048 |
| conv4 | $(256, 13, 13)$ | 43,264 | 2,048 |
| conv5 | $(256, 13, 13)$ | 43,264 | 2,048 |
| fc1 | $(4096, )$ | 4,096 | 1,024 |
| fc2 | $(4096, )$ | 4,096 | 1,024 |

**Note on noise injection.** For NMNC and VNC, noise is injected after the ReLU activation at each hidden layer listed above. The output layer is always trained with exact gradients from the cross-entropy loss.

## C. Training Details

### C.1. CIFAR-10 Training Configuration

All CIFAR-10 models were trained using stochastic gradient descent with momentum. Forward weights were optimized with a learning rate of 0.001, while feedback weights were trained with the same learning rate ($\eta_B = 0.001$). Momentum was set to 0.9, and models were trained for 100 epochs with a batch size of 64. The learning rate for forward weights was

tuned separately for each method using a logarithmic grid search. The same procedure was used for tuning the learning rate in other experiments as well.

For NMNC, incremental PCA was used to estimate low-dimensional activity manifolds online. For the default configuration, PCA bases were updated every 5 batches and the number of retained principal components per layer was set to $[512, 512, 512, 128]$.

No data augmentation was applied during training beyond standard per-channel normalization. Input images were normalized using dataset-wide means of $[0.4914, 0.4822, 0.4465]$ and standard deviations of $[0.2470, 0.2435, 0.2616]$. $\sigma_{\text{NMNC}}$ was set to 1.0.

### C.2. ImageNet Training Configuration

For ImageNet experiments, models were trained using stochastic gradient descent with momentum following (Xiao et al., 2019). Forward weights were optimized with a learning rate of 0.005, a momentum of 0.9, a weight decay of $5 \times 10^{-4}$ and a batch size of 256 for 50 epochs. A step-based learning rate schedule was applied to the forward weights, reducing the learning rate by a factor of 0.1 every 10 epochs. Feedback weights were trained using the Weight Mirror method (Akrout et al., 2019), where $B_l \leftarrow (1 - \lambda_B)B_l + \eta_B \cdot \frac{1}{N_b}\delta_l \, \delta_{l+1}^T$. To achieve reasonable performance, for both NMNC and VNC, the feedback weights were updated every batch. For NMNC, incremental PCA was updated every 10 batches (this was the highest frequency of PCA update that could keep up with model training. When the frequency was higher, the activations used for PCA accumulated leading to the out-of-memory error). As noted in (Kunin et al., 2020), the Weight Mirror method is sensitive to $\lambda_B$ and $\eta_B$, so we performed a hyperparameter search for these parameters, separately for NMNC and VNC, using Optuna (Akiba et al., 2019). Specifically, for each method, we used a tree-structured Parzen estimator with 100 sweep trials. For NMNC, the best values were $\lambda_B = 0.212$, $\eta_B = 0.101$, and for VNC, $\lambda_B = 0.414$, $\eta_B = 0.0243$. The same optimization and scheduling settings were used across all ImageNet experiments.

### C.3. RNN Training Configuration

We evaluated recurrent learning on a sequential memory task defined by parameters $(L, S, K) = (0, 5, 5)$. Each input is a sequence of integers between 0 and $K - 1$ of length $T = 2S + L$. The input presents $S$ random symbols (uniformly drawn from $\{1, \ldots, K-2\}$), followed by the blank tokens $(0)$ for $L$ steps, and a "go-cue" symbol $(K-1)$, indicating the beginning of the output timestep. The remaining $S - 1$ steps are again the blank tokens $(0)$. The target sequence is blank for the first $S + L$ steps and then reproduces the original $S$ symbols in order over the final $S$ steps, immediately upon the go-cue symbol.

We used a vanilla RNN with $H = 128$ hidden units and a $\texttt{tanh}$ nonlinearity. Models were trained for 10 epochs with batch size 256 using SGD with momentum 0.9 and learning rate $10^{-4}$ for all WP methods and $10^{-3}$ for backprop. The recurrent weights $(W_{hh})$, input weights $(W_{xh})$, and hidden bias $(b_h)$ were updated using weight perturbation, while the readout weights $(W_{hy})$ and bias $(b_y)$ were updated using exact gradients. The norm of the weight perturbation was rescaled to $\epsilon_{WP} \cdot \sqrt{NM}$, where $N$ and $M$ are the numbers of rows and columns of the weight matrix and $\epsilon_{WP} = 10^{-4}$, across different WP methods. The manifold used for perturbations was estimated online by incremental PCA on hidden states (32 PCs), updated every batch.

### C.4. Data Preprocessing

**CIFAR-10.** Images are normalized using channel-wise mean and standard deviation computed from the training set:

$$\text{mean} = [0.4914, 0.4822, 0.4465] \tag{14}$$
$$\text{std} = [0.2470, 0.2435, 0.2616] \tag{15}$$

No data augmentation is applied during training.

**ImageNet.** We use standard ImageNet preprocessing:

*Training augmentation:*

1. $\texttt{RandomResizedCrop(224)}$: Random crop with scale $(0.08, 1.0)$ and aspect ratio $(3/4, 4/3)$, resized to $224 \times 224$

2. `RandomHorizontalFlip()`: Horizontal flip with probability 0.5

3. `ToTensor()`: Convert to tensor and scale to $[0, 1]$

4. Normalization: mean $= [0.485, 0.456, 0.406]$, std $= [0.229, 0.224, 0.225]$

*Test preprocessing:*

1. `Resize(256)`: Resize shorter edge to 256 pixels

2. `CenterCrop(224)`: Center crop to $224 \times 224$

3. `ToTensor()`: Convert to tensor

4. Same normalization as training

## D. Algorithm Implementation Details

### D.1. Feedback Weight Initialization

The feedback weights $B_l \in \mathbb{R}^{n_l \times n_o}$ (where $n_l$ is the flat dimension of layer $l$'s activations and $n_o$ is the number of output units) can be initialized in two ways:

**Initial Jacobian (InitJac).** We compute the Jacobian $\frac{\partial \mathbf{y}}{\partial \mathbf{x}_l}$ at network initialization by:

1. Sampling a small batch of random inputs (batch size 32)

2. Computing the Jacobian using PyTorch's `torch.func.jacrev` and `torch.func.vmap` for vectorized computation

3. Averaging over the batch

We then initialize $B_l$ with its transpose.

**Random Initialization (No InitJac).** For comparison, we also test random initialization where the elements of the initial Jacobian are randomly permuted, destroying any meaningful gradient structure while preserving the overall statistics.

### D.2. Incremental PCA Implementation

We implement incremental PCA algorithm (Ross et al., 2008) in PyTorch for GPU acceleration. It maintains running estimates of the feature-wise mean and variance as well as the top-$k$ principal axes for $d$-dimensional activations. Each update is performed on a minibatch $X \in \mathbb{R}^{b \times d}$ (requiring $b \geq k$). The batch is centered and combined with the previous decomposition by vertically stacking (i) the prior components scaled by their singular values, (ii) the centered current batch, and (iii) a mean-correction term that accounts for changes in the running mean. An SVD of this augmented matrix yields updated principal axes and singular values; we apply the standard SVD sign-flip convention for deterministic component orientations. The algorithm returns the current principal axes as a $d \times k$ matrix (and returns random unit-norm vectors before the first update).

The SVD step can optionally use a randomized low-rank routine (`torch.svd_lowrank`); in all experiments reported here we instead compute the exact SVD via `torch.linalg.svd`, which we found faster in our setting. Finally, to keep PCA updates asynchronous without stalling training, activation minibatches are streamed to each incremental PCA instance through a bounded queue whose capacity is set by `max_queue_batches`; if this queue is full, new minibatches are dropped rather than blocking the main loop, bounding memory usage and limiting staleness of PCA updates. However, in all experiments reported here, we made sure that PCA updates were fast enough to keep up with model training without accumulating activation minibatches.

### D.3. Multiprocessing for incremental PCA

For CIFAR-10 and ImageNet experiments, NMNC training used a two-GPU multiprocessing setup to parallelize model training and incremental PCA updates:

- **Main process (GPU 0)**: Runs forward and backward passes, performs weight updates, and coordinates training.

- **PCA workers (GPU 1)**: Separate processes for each hidden layer (Conv1–3 and FC1 for CIFAR-10; Conv1–5 and FC1–2 for ImageNet), each maintaining its own incremental PCA state.

- **Communication**:
  - Activations are sent from the main process to PCA workers via per-layer multiprocessing queues.
  - Updated principal components are returned to the main process via a shared result queue.
  - PyTorch CUDA inter-process communication (IPC) is used for efficient GPU tensor transfer.

- **Non-blocking execution**: PCA updates run asynchronously with training; the main loop proceeds while workers process accumulated activations.

## E. Intrinsic Dimensionality Analysis

### E.1. Network Width Scaling

To study how manifold dimensionality scales with network size, we vary the width multiplier $n$ applied to all channel dimensions (Table 5):

*Table 5.* Network configurations for width scaling experiments.

| Multiplier $n$ | conv1 | conv2 | conv3 | fc1 | Total params (approx.) |
|---|---|---|---|---|---|
| 1/32 | 2 | 4 | 8 | 32 | 5K |
| 1/16 | 4 | 8 | 16 | 64 | 19K |
| 1/8 | 8 | 16 | 32 | 128 | 75K |
| 1/4 | 16 | 32 | 64 | 256 | 298K |
| 1/2 | 32 | 64 | 128 | 512 | 1.2M |
| 1 (default) | 64 | 128 | 256 | 1024 | 4.7M |
| 2 | 128 | 256 | 512 | 2048 | 18.8M |
| 4 | 256 | 512 | 1024 | 4096 | 75.2M |
| 8 | 512 | 1024 | 2048 | 8192 | 301M |

## F. Why low-rank perturbations do not change the fundamental variance scaling of weight perturbation

This appendix provides a short calculation supporting the statement in Footnote 4 (Section 4.5): *using low-rank perturbations can reduce the **computational** cost of each perturbation, but it does not fundamentally remove the scaling of the **gradient-estimator variance** with the number of parameters being perturbed.* The key point is that unless we restrict optimization to a lower-dimensional *parameterization* (e.g., only optimizing a low-rank factorization), the gradient being estimated still lives in a $d$-dimensional space, where $d$ is the number of perturbed parameters.

**Setup (matrix weight perturbation).** Let $W \in \mathbb{R}^{N \times M}$ be a weight matrix (e.g., $W_{hh}$ in the RNN experiments), and let $\mathcal{L}(W)$ be the scalar loss. Denote the true gradient by

$$G := \nabla_W \mathcal{L}(W) \in \mathbb{R}^{N \times M}, \qquad d := NM.$$

A common (antithetic) weight-perturbation / ES estimator uses a random perturbation $E \in \mathbb{R}^{N \times M}$ and

$$\hat{G}(E) := \frac{\mathcal{L}(W + \varepsilon E) - \mathcal{L}(W - \varepsilon E)}{2\varepsilon} E, \tag{16}$$

where $\varepsilon > 0$ controls perturbation magnitude. For $K$ i.i.d. perturbations $\{E_k\}_{k=1}^K$, we average $\hat{G}_K := \frac{1}{K} \sum_{k=1}^K \hat{G}(E_k)$.

**Small-$\varepsilon$ approximation.** For sufficiently small $\varepsilon$, a first-order Taylor expansion gives

$$\mathcal{L}(W \pm \varepsilon E) \;=\; \mathcal{L}(W) \pm \varepsilon \langle G, E \rangle + O(\varepsilon^2),$$

where $\langle A, B \rangle := \operatorname{tr}(A^\top B)$ is the Frobenius inner product. Substituting into (16) yields

$$\hat{G}(E) \;=\; \langle G, E \rangle E + O(\varepsilon^2). \tag{17}$$

**Isotropy and unbiasedness.** Write $e := \operatorname{vec}(E) \in \mathbb{R}^d$ and $g := \operatorname{vec}(G) \in \mathbb{R}^d$. If the perturbations are (second-moment) isotropic,

$$\mathbb{E}[e] = 0, \qquad \mathbb{E}[ee^\top] = I_d, \tag{18}$$

then, ignoring $O(\varepsilon^2)$ terms, (17) is unbiased:

$$\mathbb{E}[\hat{G}(E)] \;=\; \mathbb{E}[\langle G, E \rangle E] \;=\; G.$$

Importantly, the normalization used in low-rank schemes is typically chosen precisely so that (18) (or a scaled version) holds.

**Variance as Frobenius MSE scales with $d$.** A natural global measure of estimator noise is the Frobenius mean-squared error (MSE)

$$\mathbb{E}\big[\|\hat{G}_K - G\|_F^2\big] \;=\; \frac{1}{K} \mathbb{E}\big[\|\hat{G}(E) - G\|_F^2\big] \quad \text{(i.i.d. samples)}.$$

Using (17) and unbiasedness,

$$\mathbb{E}\big[\|\hat{G}(E) - G\|_F^2\big] \;=\; \mathbb{E}\big[\langle G, E \rangle^2 \|E\|_F^2\big] - \|G\|_F^2 \;+\; O(\varepsilon^2). \tag{19}$$

For many isotropic choices (including full i.i.d. Gaussian perturbations and normalized low-rank perturbations), the leading term in (19) grows linearly with $d = NM$.

**Full-rank i.i.d. Gaussian perturbations.** If $e \sim \mathcal{N}(0, I_d)$ (equivalently $E_{ij} \overset{i.i.d.}{\sim} \mathcal{N}(0,1)$), standard Gaussian fourth-moment identities imply

$$\mathbb{E}\big[\langle G, E \rangle^2 \|E\|_F^2\big] \;=\; (d+2)\|G\|_F^2,$$

and therefore

$$\mathbb{E}\big[\|\hat{G}_K - G\|_F^2\big] \;=\; \frac{d+1}{K} \|G\|_F^2 \;+\; O(\varepsilon^2). \tag{20}$$

Thus, to keep the *global* estimator noise (in Frobenius norm) constant as $d$ grows, one needs $K = \Omega(d)$ perturbation samples.

**Rank-1 perturbations (explicit calculation).** Consider rank-1 perturbations

$$E = uv^\top, \qquad u \sim \mathcal{N}(0, I_N), \;\; v \sim \mathcal{N}(0, I_M), \tag{21}$$

which correspond to $e = v \otimes u$ in vectorized form. One can verify that $\mathbb{E}[ee^\top] = I_M \otimes I_N = I_d$, so (18) holds and the estimator is unbiased (up to $O(\varepsilon^2)$).

In this case $\langle G, E \rangle = u^\top G v$ and $\|E\|_F^2 = \|u\|^2 \|v\|^2$. A direct Gaussian-moment calculation gives

$$\mathbb{E}\big[(u^\top G v)^2 \|u\|^2 \|v\|^2\big] \;=\; (N+2)(M+2)\|G\|_F^2,$$

and plugging into (19) yields

$$\mathbb{E}\big[\|\hat{G}_K - G\|_F^2\big] \;=\; \frac{(N+2)(M+2) - 1}{K} \|G\|_F^2 \;+\; O(\varepsilon^2) \;=\; \frac{d + 2N + 2M + 3}{K} \|G\|_F^2 \;+\; O(\varepsilon^2). \tag{22}$$

The leading term is still $\Theta(d/K)$, i.e., the same *dimension-driven* scaling as the full-rank Gaussian case (20), up to constants.

**Rank-$r$ low-rank perturbations.** A common rank-$r$ construction is

$$E = \frac{1}{\sqrt{r}} \sum_{k=1}^{r} u_k v_k^\top, \qquad u_k \sim \mathcal{N}(0, I_N), \ \ v_k \sim \mathcal{N}(0, I_M) \ \ \text{i.i.d.} \tag{23}$$

The $1/\sqrt{r}$ normalization ensures $\mathbb{E}[ee^\top] = I_d$ (each entry has $O(1)$ variance), so the estimator remains unbiased to first order. Increasing $r$ changes higher-order moments (and thus the *constant* in the MSE), but as long as we are still estimating a $d$-dimensional gradient over the full parameter space, the leading MSE scaling remains proportional to $d/K$.

**Interpretation.** Low-rank perturbations can be valuable because applying $E$ (or generating it) may cost $O(r(N + M))$ rather than $O(NM)$, improving computational/hardware efficiency. However, unless learning is *restricted to a lower-dimensional parameterization* (so the *unknown* gradient itself has only $O(r(N + M))$ degrees of freedom), the Monte Carlo estimator is still recovering a $d$-dimensional object. Consequently, the number of perturbations required to control the *global* estimator noise scales as $K = \Omega(d)$, and low-rank perturbations primarily affect constants rather than this fundamental scaling.

## G. TwoNN intrinsic dimension estimator

We estimate intrinsic dimensionality using the TwoNN estimator of Facco et al. (2017), which uses only the first and second nearest neighbors of each point. Given a dataset $\{x_i\}_{i=1}^{N} \subset \mathbb{R}^D$, let $r_{i,1}$ and $r_{i,2}$ denote the Euclidean distances from $x_i$ to its first and second nearest neighbors (excluding $x_i$), and define the ratio

$$\mu_i := \frac{r_{i,2}}{r_{i,1}} \in [1, \infty).$$

Under the assumption that, within the scale set by $r_{i,2}$, the sampling density is approximately constant on a locally $d$-dimensional manifold, the cumulative distribution of $\mu$ depends only on $d$ (and not on the density):

$$F(\mu) = \Pr(\mu_i \leq \mu) = 1 - \mu^{-d}, \qquad \mu \geq 1.$$

This implies the linear relation

$$-\log\bigl(1 - F(\mu)\bigr) = d \log \mu,$$

so the points $(\log \mu_{(i)}, -\log(1 - \hat{F}(\mu_{(i)})))$ lie approximately on a line through the origin with slope $d$, where $\mu_{(i)}$ are the sorted ratios and $\hat{F}(\mu_{(i)}) = (i - 0.5)/N$ is the empirical CDF. We estimate $\hat{d}$ by least-squares fitting of this line; we discard the top and bottom 10% of $\mu_{(i)}$ when fitting to reduce sensitivity to outliers and boundary effects.

## H. Brain Score evaluation

We evaluated ImageNet-trained models using the standard Brain-Score pipeline (Schrimpf et al., 2018; 2020). For each model, activations were extracted separately from each layer and linearly mapped to the corresponding neural responses for each brain area or to the behavioral response targets. Predictive performance was measured by Pearson correlation and normalized by the benchmark noise ceiling. For each Brain Score reported in Figure 6B and Table 6, we report the best-layer score for each model and benchmark.

## I. Depth-scaling of neural manifold dimensionality

For the depth-scaling analysis, we constructed CIFAR-10 networks by repeating each convolutional stage while preserving the downsampling locations of the default architecture. Additional layers within a stage used stride 1, so the spatial resolution at the end of each stage matched the default network. We varied width and depth multipliers jointly, trained each configuration with the same CIFAR-10 recipe, and measured manifold dimensionality at the final layer of each stage (Conv1–Conv3) and at FC1. Dimensionality was estimated using both TwoNN and the number of PCs required to explain 90% activation variance, with mean $\pm$ std reported across $n = 5$ seeds.

# J. Supplementary figures

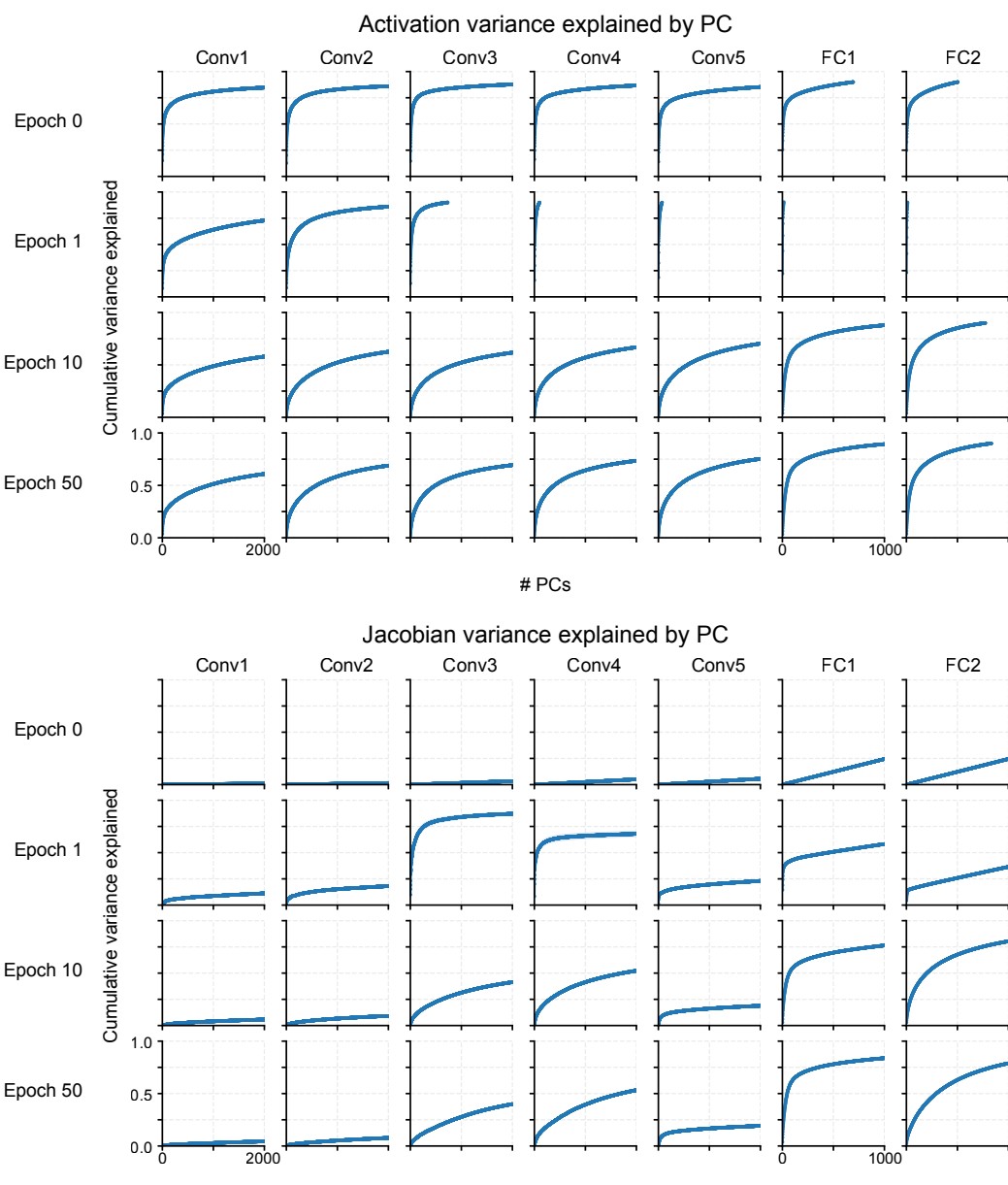

*Figure 8.* Same as Figure 1 for the ImageNet model. Note that the x-axis runs from 0 to 2000 for Conv1-5 and from 0 to 1000 for FC1-2.

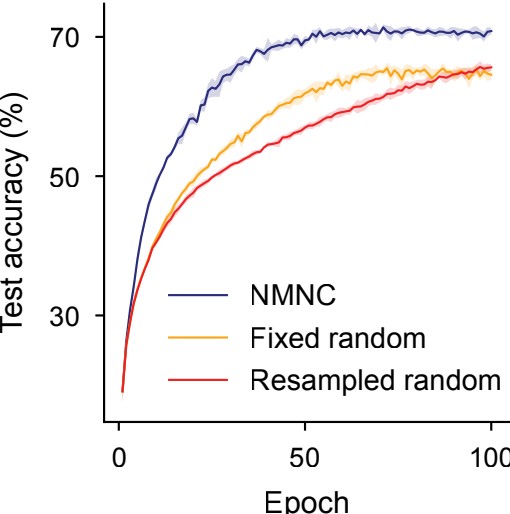

*Figure 9.* Test accuracy vs. epochs for random subspace controls of NMNC. "Fixed random" and "Resampled random" refer to learning rules identical to NMNC except that the noise is sampled from a fixed or resampled (every time we inject noise) random subspace of the same dimensionality as the neural manifold used in NMNC. The learning rules shown here are all No-InitJac variants.

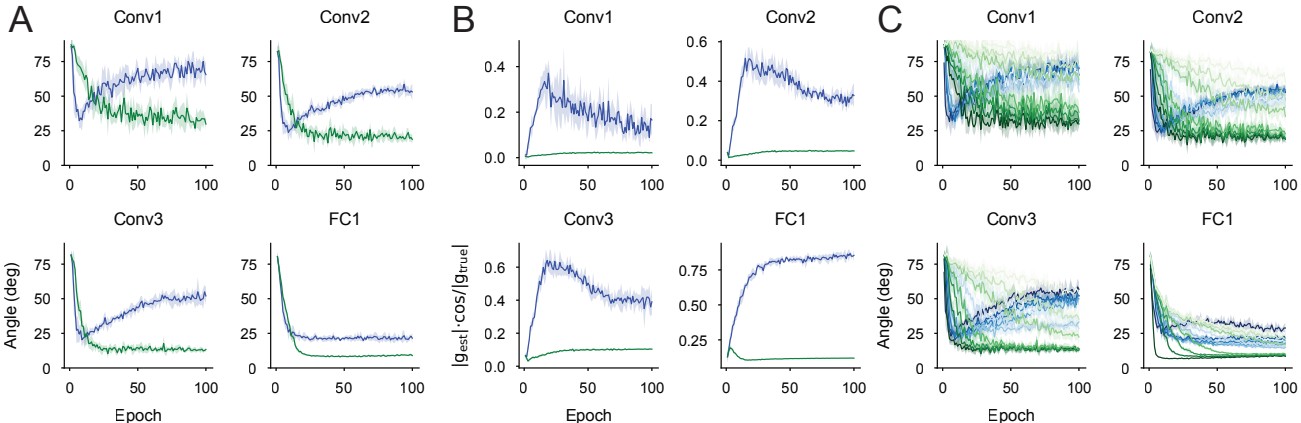

*Figure 10.* Same as Figure 4 for alignment in weight space.

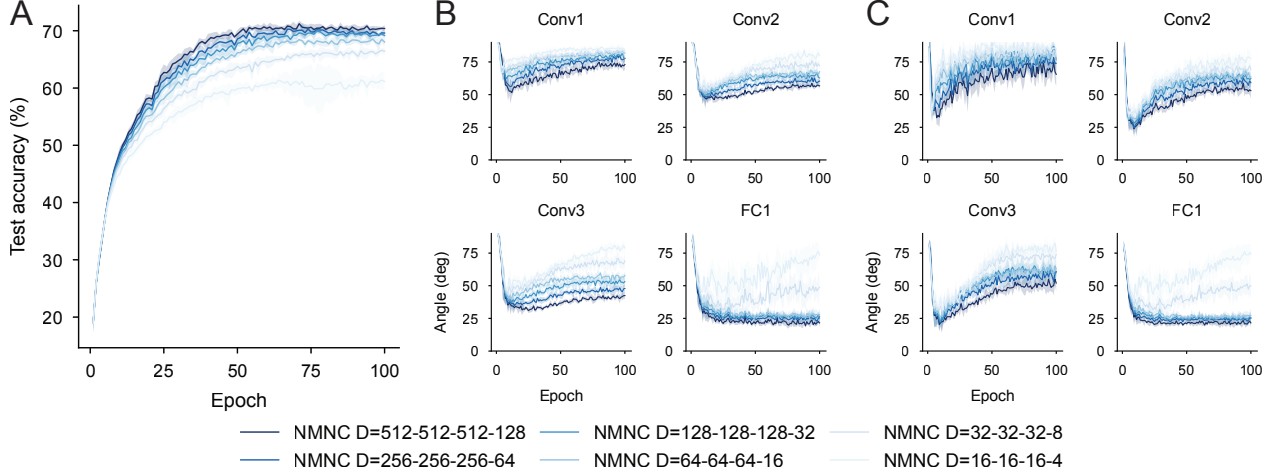

*Figure 11.* NMNC with varying numbers of neural manifold dimensions (PCs) for the CIFAR-10 model. (A) Test accuracy vs. epochs. (B) Alignment between true and estimated gradients in activation space. (C) Alignment in weight space.

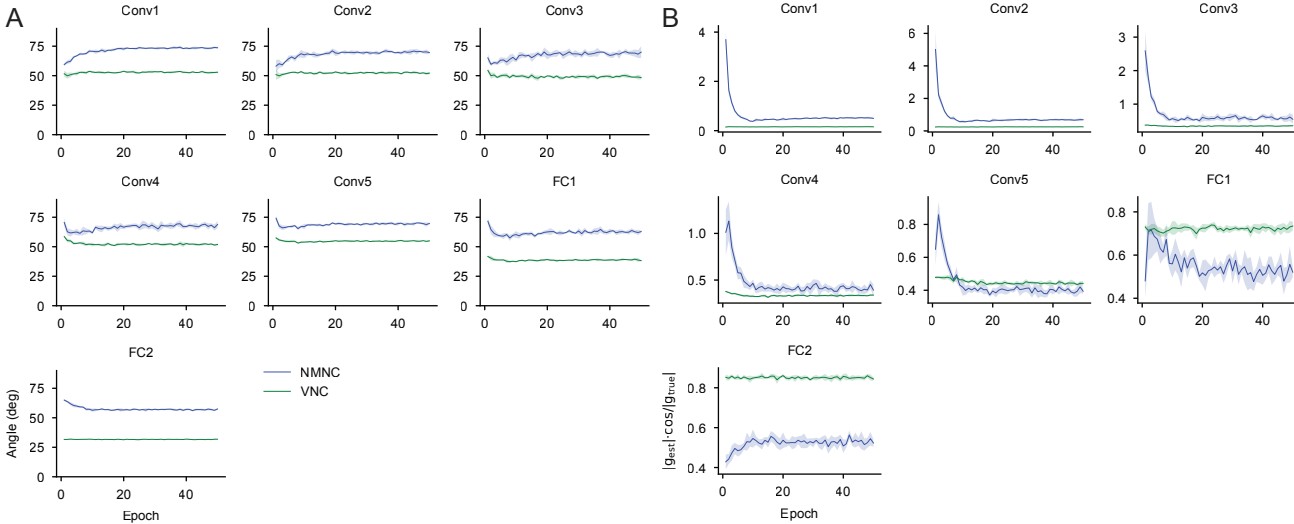

*Figure 12.* Alignment between true and estimated gradients in activation space for the ImageNet model. (A) Angle between the true and estimated gradients for NMNC and VNC across layers. (B) Normalized magnitude of the estimated gradient projected onto the true gradient direction for NMNC and VNC across layers.

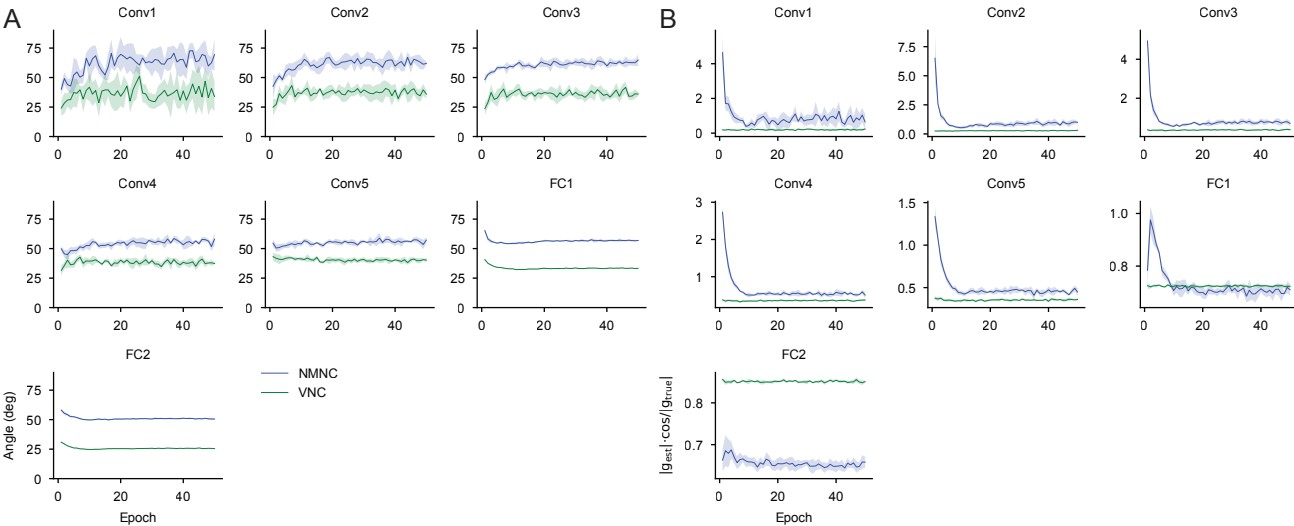

*Figure 13.* Same as Figure 12 for alignment in weight space.

## K. Supplementary tables

*Table 6.* Performance-matched comparison of neural representations from ImageNet-scale models trained with NMNC and VNC. Scores are mean ± std across $n = 5$ seeds. "Performance-matched NMNC" refers to epoch-11 NMNC checkpoints whose ImageNet accuracy matches that of the VNC-trained models. High-frequency DFT refers to the discrete Fourier transform of the Conv1 kernels averaged over $f > 0.3$

| Metric | NMNC | Performance-matched NMNC | VNC |
|---|---|---|---|
| V4 Brain Score | $0.519 \pm 0.006$ | $0.497 \pm 0.002$ | $0.475 \pm 0.001$ |
| IT Brain Score | $0.457 \pm 0.003$ | $0.438 \pm 0.012$ | $0.423 \pm 0.006$ |
| Behavior Brain Score | $0.169 \pm 0.007$ | $0.162 \pm 0.006$ | $0.127 \pm 0.017$ |
| High-frequency DFT ($\times 10^{-3}$) | $2.93 \pm 0.24$ | $2.95 \pm 0.20$ | $5.96 \pm 0.04$ |
| ImageNet test accuracy (%) | $39.0 \pm 0.3$ | $34.5 \pm 0.4$ | $35.0 \pm 0.2$ |

*Table 7.* Width and depth scaling of manifold dimensionality across layers in the CIFAR-10 model. Values are mean ± std across $n = 5$ seeds. Width and depth multipliers are relative to the default architecture. PCA reports the number of PCs required to explain 90% activation variance; TwoNN reports intrinsic dimension.

| Layer / estimator | Depth \ width | 1× | 2× | 3× |
|---|---|---|---|---|
| **Conv1** | | | | |
| TwoNN | 3× | $59.8 \pm 5.2$ | $66.6 \pm 4.1$ | $62.3 \pm 2.0$ |
| | 2× | $39.5 \pm 4.0$ | $39.5 \pm 5.3$ | $37.2 \pm 1.9$ |
| | 1× | $37.9 \pm 0.8$ | $39.4 \pm 0.7$ | $39.7 \pm 0.7$ |
| PCA | 3× | $2705.0 \pm 245.3$ | $3915.8 \pm 118.0$ | $4092.0 \pm 8.9$ |
| | 2× | $720.6 \pm 334.6$ | $929.8 \pm 532.9$ | $711.0 \pm 195.2$ |
| | 1× | $364.8 \pm 23.3$ | $435.2 \pm 21.0$ | $471.8 \pm 18.7$ |
| **Conv2** | | | | |
| TwoNN | 3× | $12.9 \pm 0.4$ | $12.6 \pm 0.7$ | $11.7 \pm 0.3$ |
| | 2× | $21.3 \pm 0.9$ | $21.5 \pm 0.5$ | $21.5 \pm 0.6$ |
| | 1× | $27.7 \pm 0.4$ | $28.5 \pm 0.4$ | $29.0 \pm 0.3$ |
| PCA | 3× | $6.2 \pm 0.4$ | $5.8 \pm 0.8$ | $5.6 \pm 0.5$ |
| | 2× | $38.2 \pm 4.3$ | $41.0 \pm 3.8$ | $41.0 \pm 2.7$ |
| | 1× | $123.6 \pm 7.7$ | $139.8 \pm 7.1$ | $153.2 \pm 4.9$ |
| **Conv3** | | | | |
| TwoNN | 3× | $9.4 \pm 0.5$ | $8.9 \pm 0.2$ | $8.6 \pm 0.1$ |
| | 2× | $21.9 \pm 0.9$ | $22.0 \pm 0.9$ | $22.1 \pm 0.6$ |
| | 1× | $26.7 \pm 0.6$ | $27.4 \pm 0.6$ | $27.6 \pm 0.3$ |
| PCA | 3× | $7.2 \pm 1.1$ | $5.8 \pm 0.8$ | $6.0 \pm 0.7$ |
| | 2× | $254.6 \pm 52.0$ | $285.8 \pm 74.0$ | $302.6 \pm 38.8$ |
| | 1× | $220.2 \pm 10.8$ | $268.2 \pm 29.7$ | $300.0 \pm 18.1$ |
| **FC1** | | | | |
| TwoNN | 3× | $7.6 \pm 0.5$ | $7.2 \pm 0.3$ | $7.0 \pm 0.2$ |
| | 2× | $20.2 \pm 0.7$ | $20.5 \pm 1.0$ | $20.6 \pm 0.7$ |
| | 1× | $22.7 \pm 0.6$ | $22.9 \pm 0.7$ | $23.2 \pm 0.4$ |
| PCA | 3× | $7.2 \pm 0.4$ | $7.0 \pm 0.0$ | $6.6 \pm 0.5$ |
| | 2× | $240.6 \pm 44.1$ | $305.0 \pm 89.2$ | $342.0 \pm 48.5$ |
| | 1× | $261.8 \pm 32.8$ | $361.8 \pm 43.4$ | $431.4 \pm 28.8$ |

