# OpenReview forum: "Credit Assignment via Neural Manifold Noise Correlation"
_ICML.cc/2026/Conference — ICML 2026 regular_

### Official Review · Reviewer_ELYX · 2026-02-25

**Soundness:** 3
**Presentation:** 3
**Significance:** 2
**Originality:** 3
**Overall Recommendation:** 5
**Confidence:** 4

**Summary:**

The paper proposes a new brain-inspired learning algorithm called neural manifold noise correlation (NMNC). NMNC builds on noise correlation-based methods, which locally estimate gradients by correlating perturbations of activities or weights with changes in network predictions. However, these methods scale poorly with model size, and neural activity is known to lie on a low-dimensional manifold (the “neural manifold”). NMNC aims to address these issues by restricting perturbations to the neural manifold.

The authors provide both theoretical arguments and experiments for the claim that the dependence of the network output on the hidden activations (specifically the row space of the network Jacobian) evolves, with training, to lie on a lower dimensional manifold than the full activation space. Experiments are also provided to suggest that the dimensionality of the neural manifold scales favourably with model width.

NMNC is evaluated on convolutional networks trained on CIFAR-10 and ImageNet, suggesting better performance and alignment with the true (backpropagation, BP) gradient compared to vanilla noise correlation. On ImageNet, the kernels of the first convolutional layer trained with NMNC are shown to resemble, as for BP, Gabor filters, but those of vanilla noise correlation show a bias towards high-frequency components. The representations learned by NMNC also appear to be more predictive of primate ventral visual representations than vanilla noise correlation, as assessed on Brain Score benchmarks.

Finally, NMCN is applied to RNN in the form of weight perturbations. Results suggest that restricting perturbations to the neural manifold leads to better performance and gradient alignment, compared to a range of baselines incrementally similar to NMCN.

The authors conclude by discussing the potential implications of their findings for neuroscience and important limitations of their work.

**Compliance With Llm Reviewing Policy:**

Affirmed.

**Final Justification:**

I recommend clear acceptance (5) because, following the rebuttal, I think that overall the paper is sound, original, clearly presented, and fair in significance. The rebuttal addressed all of my concerns more than satisfactorily, which were related to potential methodological issues and validation of theoretical claims. This increased my confidence in the soundness of the work, and so I increased my initial score to a 5.

**Key Questions For Authors:**

For main questions and comments, see the major weaknesses above. I have some other minor questions:
* Why are curves shown up to 90% variance in Figure 1?
* While the work mainly builds on noise-correlation methods as bio-plausible algorithms, it would be interesting to know what the authors think about the relationship between this algorithm and other local learning rules, such as equilibrium propagation, forward learning, predictive coding, etc.

**Limitations:**

As noted in the strengths above, the authors discuss a variety of limitations of their proposed method in a satisfactory manner.

**Strengths And Weaknesses:**

**Strengths**
* The paper is overall well written and structured.
* Very clear motivation, highlighting the issues with vanilla noise correlation methods (from both a computational and neuroscience perspective).
* Clear presentation of the proposed method (NMCN) and comparison to vanilla noise correlation methods.
* Good synergy of theoretical and empirical results.
* Evaluation and application of the proposed algorithm on qualitatively different datasets (small vs large scale), models (convolutional networks vs RNNs), and tasks (e.g. classification vs Brain Score).
* Satisfactory discussion of limitations and related future directions.

**Weaknesses**

*Major*
* **Untuned learning rates**: From the experimental details in the appendix, none of the experiments seem to properly control for the learning rate across different algorithms (e.g. NMNC vs VNC) by performing standard grid searches, and this decision is not justified. This is a fundamental confounder that potentially calls into question all the empirical results, especially those related to Figure 3, 4, 5 and 7. On CIFAR-10, for example, the slower convergence and smaller gradient projections of VNC makes one wonder whether a higher learning rate would impact the results. I strongly encourage the authors to either provide a proper justification for this decision or rerun all the experiments by tuning the learning rate for each algorithm.
* **Unclear validation of theoretical claim**: It is not entirely clear from the results of Figure 1 whether the Jacobian aligns with the neural manifold, as even at the end of training a large number of principal components (PCs) seem to be necessary to account for most of the Jacobian variance. In a footnote, the authors provide two possible explanations for why the activation variance is much lower dimensional: (i) their use of PCA on the current (as opposed to the whole history of) activations; and (ii) the impact of downstream weights and therefore activations and errors. While (i) seems to be a methodological issue that could be addressed - and would be useful to see addressed - (ii) appears to be much more fundamental, and so the validity of the theoretical claim that the Jacobian lies on a lower dimensional manifold remains unclear.
* **Unaddressed scalability of neural manifold dimensionality with network depth**: The authors implicitly use model width or number of hidden units as a proxy for network size. However, depth is arguably equally important. The results of Figure 2 suggest that earlier layers have have much higher intrinsic dimensionality, begging the question of what one would see at higher depths. Overall, it would be very useful to extend the results of Figure 2 to depth - perhaps in the form of a heatmap with width and depth as axes - to better understand the full scalability of the neural manifold.

*Minor*
* No details seem to be provided on how the representations of the models trained on ImageNet were evaluated on the Brain Score.
* Many paragraphs end on a new line with one or a few words, and so space could be better optimised.

**Summary**

Overall, this is a high-quality piece of work proposing an interesting new algorithm that aims to address clear limitations of previous vanilla noise correlation methods. The method is clearly motivated from both a computational and neuroscience perspective and extensively evaluated on a number of datasets, tasks and settings. However, the experiments seem to be fatally confounded by the lack of control for learning rate across algorithms, calling into question the validity of many of the conclusions. The empirical validity of some of the theoretical claims also appears not entirely clear, and the scalability of the neural manifold dimensionality with the depth should also be investigated. I would be happy to increase my score if all these points are properly addressed.

---

> ### Author Rebuttal · Authors · 2026-03-29
>
> > ***Untuned learning rates***
>
> We thank the reviewer for raising this point. We did tune the learning rate separately for each method using a logarithmic grid search. On CIFAR-10, we performed a search over {1e-2, 1e-3, 1e-4, 1e-5} and found 1e-3 to be best for all methods  (1e-2 caused divergence); we applied the same tuning procedure for all other experiments. We will state this clearly in the revision.
>
> > ***Unclear validation of theoretical claim***
>
> We thank the reviewer for the careful reading. On the first point, we agree that using only current activations in PCA could be a concern. We therefore repeated the Fig. 1 analysis with incremental PCA over the activation history. The result is similar to before, with a modest improvement as expected; e.g., at epoch 100, Jacobian variance explained by the top 500 PCs was 0.58/0.53/0.33/0.88 when using the current activations, and 0.61/0.53/0.35/0.92 when using the activation history for Conv1/2/3/FC1. We will add this result to the revision.
>
> On the second point, we agree this is more fundamental: differential weighting by downstream weights and errors (which depend on task, architecture, and training history) prevents a precise theoretical characterization. We therefore rely on empirical validation: Fig. 1 shows that the neural manifold captures Jacobian variance substantially better than a random subspace of equal dimension (which would yield linear cumulative variance, cf. epoch 0). As additional support, we repeated this analysis for ImageNet: before training, Jacobian variance explained was at a chance level (0.01–0.06 for Conv1–5 with 2000 PCs; 0.24/0.24 for FC1/2 with 1000 PCs), but increased substantially after training began (epoch 1: 0.11/0.18/0.87/0.68/0.23 for Conv1–5; 0.58/0.36 for FC1/2), confirming that the alignment emerges with training and generalizes beyond CIFAR-10. Accordingly, we will soften the claim: our results support preferential alignment of the Jacobian with the neural manifold, rather than full containment.
>
> > ***Unaddressed scalability of neural manifold dimensionality***
>
> We agree that depth is an important scaling axis. We focused on width because neural manifolds are naturally defined layer-wise (reflecting features at a specific level of abstraction) and scaling width directly probes representational capacity within a layer.
>
> Scaling depth introduces potential confounds: (i) beyond a sufficient depth, networks may propagate or reuse representations, artificially inflating total dimensionality. (ii) multiple valid depth-scaling schemes exist, leading to interpretive ambiguity.
>
> Nevertheless, following the reviewer’s suggestion, we performed a controlled depth-scaling experiment by repeating each conv stage while preserving downsampling locations (extra layers use stride 1; e.g., Conv1 → Conv1 (stride 1) → Conv2 → Conv2 (stride 1) → Conv3 → Conv3 (stride 1) → FC1, for a 2x depth scaling). We measured manifold dimensionality at the end of each stage and FC1.
>
> We found that dimensionality increases with depth only in Conv1, but decreases in later layers, consistent across PCA and TwoNN. In the table below, we show the result of simultaneous width and depth scaling for FC1 using the TwoNN method in a heat-map like format, as the reviewer suggested (mean ± std, n=5 seeds). Importantly, test accuracy remained stable across settings (~70%), suggesting this is not due to optimization failure.
>
> These results indicate that, beyond early layers, increasing depth promotes dimensionality compression rather than growth, complementing our width-scaling results.
>
> | depth↓\width→ | 1           | 2           | 3           |
> |:-------------:|:-------------:|:-------------:|:-------------:|
> | **3**           | 7.6 ± 0.5   | 7.2 ± 0.3   | 7.0 ± 0.2   |
> | **2**           | 20.2 ± 0.7  | 20.5 ± 1.0  | 20.6 ± 0.7  |
> | **1**           | 22.7 ± 0.6  | 22.9 ± 0.7  | 23.2 ± 0.4  |
>
> > **Minor**
> * **Brain-Score details**. We followed the standard Brain-Score pipeline. Model activations (per layer) were linearly mapped to neural responses for each brain area or behavioral responses, and performance was measured using noise-ceiling-normalized Pearson correlation. We reported the result for the best layer per model for each score. We will clarify this in the revision.
> * **Formatting/space**. Thank you for noting this. We will revise paragraph formatting to improve space efficiency.
>
> > **Minor questions**
> * **Why 90% variance in Fig. 1?** This follows common practice in neuroscience, where manifold dimensionality is defined via a fixed variance threshold. We chose 90% to provide an interpretable and consistent reference point.
> * **Relation to other local learning rules.** Equilibrium propagation and predictive coding differ from NMNC in that they approximate gradients through recurrent relaxation/inference dynamics. Forward learning is also distinct from NMNC as it uses layer-local objectives instead of end-to-end gradient estimation.

---

> > ### Author Rebuttal · Reviewer_ELYX · 2026-04-01
> >
> > I thank the authors for their clear and thorough response. I selected **(a)** because the rebuttal addresses all of my concerns, related to the soundness (rather than originality or presentation of the work). In particular, I appreciate:
> > * the clarification of tuning of learning rates,
> > * the discussion on the dimensionality of the network Jacobian and additional supporting experiments, and
> > * the additional experiment testing the impact of the network's aspect ratio (width vs depth) on manifold dimensionality. It is very interesting that for most layers the dimensionality tends to decrease with the depth. As a suggestion, these results could be added to the appendix and referred to in the main text when discussing the results of Figure 2.
> >
> > These points clarify and strengthen the soundness of the work. As agreed by the authors, the revised version of the manuscript should clarify and qualify the theoretical claims about the dimensionality of the network Jacobian, in line with feedback from other reviewers.
> >
> > Beyond this, I am glad that the authors addressed the above concerns, since I think this is a very good example of meaningful work at the intersection between neuroscience and AI which opens up interesting future directions. I will therefore increase my score.

---

> > > ### Author Response · Authors · 2026-04-06
> > >
> > > We thank the reviewer for the thoughtful feedback and for appreciating our work. We will revise the manuscript to reflect the clarifications discussed in the rebuttal, and will include the additional results on width–depth scaling of manifold dimensionality in the revised version.

---

### Official Review · Reviewer_9Pun · 2026-03-11

**Soundness:** 3
**Presentation:** 4
**Significance:** 3
**Originality:** 3
**Overall Recommendation:** 5
**Confidence:** 3

**Summary:**

The paper presents Neural Manifold Noise Correlation (NMNC), a noise correlation-based solution to the credit assignment problem. NMNC aims to overcome the scaling problem that vanilla noise correlation methods experience with increasing network size by estimating the Jacobian with noise perturbations on the low-dimensional neural manifold instead of utilizing isotropic noise. From a theoretical perspective, this approach is motivated by the observation that the Jacobian row space aligns with the neural manifold. That is, only the noise components that are parallel to the neural manifold probe meaningful input-output relationships for estimating the Jacobian as downstream layers in the network are trained only on the activations of preceding layers. The paper confirms this theoretical consideration empirically by showing that the Jacobian of a CIFAR-10 network trained by backpropagation aligns with the (PCA-defined) neural manifold.

To learn feedback weights, NMNC performs noise correlation within each layer’s activity manifold. For each layer, a low-dimensional basis is estimated online from neural activations using incremental PCA. Noise is sampled in the low-dimensional manifold and projected to the full space via this basis, instead of sampling isotropic noise in the full activation space as is the case in vanilla noise correlation.

The paper evaluates the method in three experiments: (i) direct-feedback learning in a CNN trained on CIFAR-10; (ii) layerwise feedback training of AlexNet on ImageNet, and (iii) weight perturbation training for recurrent networks.

(i) On CIFAR-10, training the CNN with NMNC performed substantially better than training with vanilla noise correlation (VNC) and direct feedback alignment (DFA). Training with backpropagation remained the best performing method overall. NMNC also had a better sample efficiency than VNC and DFA. Comparing the pseudo-gradients to the true backpropagation gradients showed that NMNC yielded a better alignment in lower layers early in training and made larger effective steps along the true gradient.
(ii) On ImageNet, training AlexNet with a layerwise feedback regime and NMNC outperformed training with VNC. Backpropagation remained significantly better. While VNC exhibited better alignment in this scenario, NMNC had larger pseudo-gradient magnitudes and projected steps. The paper also evaluated the effect of NMNC on the interal representations of AlexNet, showing that VNC experiences yields kernels with strong high-frequency components (resulting in salt-and-pepper patterns) while NMNC representations are not affected by such high-frequency patterns. Comparing the learned representations to representations in the primate visual stream showed that backpropagation obtained the highest BrainScore, followed by NMNC and then VNC.
(iii) The paper tested whether restricting weight perturbations (WP) to the neural manifold during training of an RNN performs better than other WP methods. On a sequential memory task, the manifold-based WP approach (rank-1 manifold WP) achieved the best performance and showed higher gradient alignment and larger pseuo-gradient magnitude.

The paper concludes that NMNC can reduce the effective dimensionality of credit assignment, increasing the practicality of noise correlation-based methods for biologically plausible credit assignment. Limitations that are identified include the online, incremental PCA approach, which can (1) only provide a linear subspace approximation, (2) is computationally expensive, and (3) is not biologically plausible. The paper also identifies research directions to replace the online PCA with more suitable alternatives.

**Compliance With Llm Reviewing Policy:**

Affirmed.

**Final Justification:**

The authors satisfactorily addressed my comments. In view of this and responses to the other reviewers I updated my score to accept.

**Key Questions For Authors:**

Can you explain the discrepancy in alignment results between the CIFAR-10 implementation and the AlexNet implementation? Further, can you explain why you consider the strong early alignment an important factor for the success of the NMNC implementation on CIFAR-10 even though the NMNC implementation on ImageNet shows that a substantial misalignment does not reduce the performance of NMNC in comparison to VNC?

What is the explanation for the increasing misalignment between the true and estimated gradient for the recurrent network implementation of NMNC (WP Rank 1, Figure 7 B) and what are the expected effects of this trend after more gradient alignment steps? This approach also exhibits substantially more variance in terms of alignment angle and estimated gradient magnitude. Could this variance be reduced by more optimal manifold estimation techniques or is this caused by a different phenomenon?

**Limitations:**

Yes

**Strengths And Weaknesses:**

Soundness: The claims are supported by theoretical analysis as well as empirical experiments. The paper also investigates the mechanisms underlying the results, but here the conclusions are less clear. While the paper argues that part of the reason why NMNC outperforms VNC on CIFAR10 is the better early gradient alignment, yet NMNC shows substantially larger misalignment in the AlexNet implementation tested on ImageNet. This contradiction seems to question the impact of (early) gradient alignment and instead points towards a more prominent role for the step size along the true gradient (as this is consistently larger in both cases). However, the paper does not address this discrepancy. Further, weaknesses and limitations are identified in the Discussion, but not empirically verified or quantified (e.g. computational cost of online incremental PCA for AlexNet).
Presentation: The manuscript is very well written and structured, combining theoretical and empirical observations in a clear narrative. Figures are clear and intuitive.
Significance: Biologically plausible credit assignment, the topic of the present paper, remains a crucial yet unresolved question in the field of machine learning. The proposed approach investigates the suitability of neural manifold noise correlation to improve performance and efficiency of noise based-correlation methods. By evaluating NMNC across various learning settings, a deeper understanding of the effectiveness of NMNC is provided, strengthening the impact of the work. Although the approach still suffers from several weaknesses, such as the computational cost (making practical applications for large input activations prohibitively expensive) and biological implausibility of online PCA to estimate the manifold, the paper contributes relevant insights regarding the potential of such approaches and identifies new research directions. The investigation of the impact of the noise correlation-based methods (including NMNC) on internal representations also advances understanding in the field.
Originality: Proposing Neural Manifold Noise Correlation, the paper provides original insights into efficient noise perturbation strategies and more biologically plausible noise correlation-based solutions to the credit assignment problem.

---

> ### Author Rebuttal · Authors · 2026-03-29
>
> > **Can you explain the discrepancy in alignment results between the CIFAR-10 implementation and the AlexNet implementation? Further, can you explain why you consider the strong early alignment an important factor for the success of the NMNC implementation on CIFAR-10 even though the NMNC implementation on ImageNet shows that a substantial misalignment does not reduce the performance of NMNC in comparison to VNC?**
>
> We thank the reviewer for raising this important point. We agree that our current wording overstates the role of alignment. The CIFAR-10 and ImageNet experiments are in different regimes, so the same mechanism need not dominate in both.
>
> In CIFAR-10, NMNC is used in a direct-feedback setting with large output-to-hidden feedback matrices, and feedback is updated only every $b$ batches ($b=5$ in the main experiments). In this sample-limited regime, restricting perturbations from the full $n_l$-dimensional activity space to a $d_l$-dimensional manifold gives NMNC a strong early estimation advantage: Eq. (12) predicts better early cosine alignment when $\alpha_l > d_l/n_l$, which is the regime suggested by Fig. 1, and Fig. 4 shows that this advantage is largest early in training and in lower, higher-dimensional layers. We therefore view early alignment as an important mechanism for the CIFAR-10 result.
>
> By contrast, the ImageNet experiment uses layerwise Weight Mirror because direct feedback was unstable (likely because the linear approximation of the Jacobian is not good enough at this scale). In that setting, the feedback objects being learned, the transposed kernels, are much lower-dimensional than in the CIFAR-10 direct-feedback regime, and feedback is updated every batch. This makes VNC less sample-limited than in CIFAR-10, so it can match or exceed NMNC in cosine alignment. However, this does not negate NMNC’s advantage: Eq. (13) and the argument following it shows that manifold-restricted perturbations can still produce a larger component along the true gradient, and this is what we observe in Fig. 10B, especially in the convolutional layers. We will revise the text to say that early alignment is an important mechanism in the CIFAR-10 direct-feedback regime, but not a universal necessary condition for NMNC to outperform VNC. On ImageNet, the larger projected step along the true gradient appears to be the more relevant mechanism.
>
> > **What is the explanation for the increasing misalignment between the true and estimated gradient for the recurrent network implementation of NMNC (WP Rank 1, Figure 7 B) and what are the expected effects of this trend after more gradient alignment steps? This approach also exhibits substantially more variance in terms of alignment angle and estimated gradient magnitude. Could this variance be reduced by more optimal manifold estimation techniques or is this caused by a different phenomenon?**
>
> We thank the reviewer for pointing out these interesting phenomena.
>
> First, the increasing misalignment for NMNC ("WP rank 1 (manifold)") may arise from limitations of the online manifold estimate. In our implementation, the perturbation subspace is updated online via incremental PCA on hidden states collected throughout training. As training converges, the cumulative PCA basis may become dominated by the later, more stable hidden-state distribution and under-represent older directions that still matter for the Jacobian, so the perturbation subspace can gradually miss relevant gradient components. If that trend continues, we would expect the projected component along the true gradient to decrease, which could slow learning and produce a larger asymptotic gap to backpropagation, although within the training window of Fig. 7 the manifold method still performs best among the WP variants. We believe that a more refined approach that addresses the under-weighting of earlier activations could mitigate the increasing misalignment for NMNC.
>
> We would also like to note that the slower increase in misalignment shared by all methods may reflect a more general limitation of one-sided finite-difference gradient estimation: as training progresses and the true gradient becomes smaller, the higher-order terms omitted by the finite-difference approximation can become relatively more influential, leading to worse angular alignment even if the estimator remains useful for learning.
>
> Regarding the higher variance of NMNC compared to other methods in the recurrent setting, we do think that this is caused by stochasticity in the online manifold estimation, which can cause the perturbation directions to fluctuate over training, and may be mitigated by improved manifold estimation techniques.

---

> > ### Author Rebuttal · Reviewer_9Pun · 2026-04-02
> >
> > The authors satisfactorily addressed my comments. In view of this and responses to the other reviewers I updated my score to accept.

---

> > > ### Author Response · Authors · 2026-04-06
> > >
> > > We thank the reviewer again for the insightful questions and the positive assessment. We are glad that our responses clarified the points raised.

---

### Official Review · Reviewer_mbXq · 2026-03-12

**Soundness:** 1
**Presentation:** 2
**Significance:** 3
**Originality:** 3
**Overall Recommendation:** 2
**Confidence:** 4

**Summary:**

This paper studies perturbation-based credit assignment, and proposes neural manifold noise correlation (NMNC), a variant of noise correlation that restricts perturbations to a low-dimensional subspace which is estimated online from hidden activations using incremental PCA. The paper motivates the method by arguing, both empirically and mathematically, that in trained networks the Jacobian row space is largely aligned with the neural activity manifold, and that this manifold dimension grows much more slowly than raw layer width. The results suggest NMNC improves over vanilla noise correlation on a CIFAR-10 convolutional network, an AlexNet/ImageNet setting with layerwise feedback, and a recurrent memory task, and it also yields somewhat more brain-like visual representations than VNC in the ImageNet experiment.

**Compliance With Llm Reviewing Policy:**

Affirmed.

**Key Questions For Authors:**

1. In the feedforward experiments, did the authors test a random low-dimensional subspace baseline with the same dimension as NMNC? If yes, please report it. If no, can the authors explain why that control was omitted despite including an analogous fixed-subspace baseline in Figure 7A for the RNN case?
2. How were the default manifold dimensions in Table 2 and Table 4 chosen? Did the authors test if the method is robust across a border range of d_I?
3. Can the authors provide any wall-clock time or memory measurements for NMNC versus VNC, particularly for AlexNet experiments, to justify the sample efficiency and scalability?

**Limitations:**

Yes.

**Strengths And Weaknesses:**

Strengths :
1. The central idea is straightforward and interesting: if the useful gradient information mostly lives in a low-dimensional activity subspace, then perturbing only in that subspace should reduce estimator variance. This is a clean, novel contribution to the field.
2. The motivations are visually well-presented and justified. Figure 1 clearly shows both activation variance and Jacobian variance becoming concentrated in relatively few PCs during training on CIFAR-10. Figure 2 presents the other claim, that the effective dimensionality is much smaller than the ambient dimensionality.
3. The experiments are set on CIFAR-10, ImageNet-scale setup, and an RNN task, which represents a broader scope, especially compared with many other papers in biologically plausible learning, which normally carried out in toy setting.

Weaknesses:
1. In the feedforward experiments, NMNC is compared against full-space VNC, but not against a random subspace baseline of the same dimension as used in NMNC. Hence the results presented in Figure 3A-C and Figure 5 is open to a simple explanation that any sufficiently low-dimensional perturbation scheme would improve estimator variance. Some further control experiments will justify the findings better.
2. The paper claims the “scalability” of NMNC, however, the authors also explain that they use AlexNet because incremental PCA is too expensive for more modern architectures with very large activations, e.g. table 4 shows conv1 has 193,600 dimensions even before any reduction. The appendix indicates a two-GPU asynchronous PCA setup. Taken together, the method can be made to run on AlexNet/ImageNet, but not yet that it is a scalable solution in a broader sense. In addition, the paper does not provide any runtime and memory numbers, hence it is hard to infer the scalability.
3. The key hyperparameter dependence is not discussed in the main paper. NMNC’s core idea comes from trading ambient dimension for a reduced subspace, hence the performance could depend heavily on the chosen manifold dimension d_l. Though d_l are reported in Table 2 and 4 for each experiments, the main paper does not justify the choice of those numbers.
4. The brain-score evaluation may not lead to a clear conclusion.  In Figure 6B, NMNC sits between bp and VNC on brain score, but the same ordering also holds for task accuracy, hence the brain score result is not clearly disentangled from general model quality. The paper claims that HMNC yields more brain-like visual representations than VNC may therefore need further justifications.
5. The surrounding algorithmic setup changed for CIFAR-10 and ImageNet. On CIFAR-10 the paper studies direct-feedback learning. On ImageNet, direct feedback apparently became unstable, so the authors switched to a layerwise Weight Mirror scheme. This is not invalid, but it means the paper is demonstrating a perturbation principle that can be inserted into multiple frameworks, not one stable credit-assignment method across scales, which is a somehow weaker and more qualified claim than the introduction suggests.

---

> ### Author Rebuttal · Authors · 2026-03-29
>
> > **Random subspace control**
>
> We thank the reviewer for mentioning this important control. We actually conducted this control for the CIFAR-10 experiment but decided not to include it because we thought that focusing on the comparison between NMNC and VNC would be conceptually clearer when we first introduce these methods. Below are the test accuracies across epochs for the fixed and resampled (every time we inject noise) random subspace model, VNC, and NMNC (the same analysis as Fig. 3A). We found that both random subspace models exhibited similar performance curves to VNC (mean ± std, n=5 seeds).
> | Model ↓ / Accuracy (%) → | Epoch 1 | Epoch 25 | Epoch 50 | Epoch 100 | Best test acc |
> | :---: | :---: | :---: | :---: | :---: | :---: |
> | Fixed random | 19.1 ± 1.5 | 52.4 ± 0.3 | 61.9 ± 0.6 | 64.6 ± 0.6 | 66.2 ± 0.3 |
> | Resampled random | 19.1 ± 1.6 | 49.8 ± 0.4 | 57.0 ± 0.4 | 65.6 ± 0.5 | 65.9 ± 0.6 |
> | VNC | 18.6 ± 1.7 | 50.6 ± 0.4 | 58.2 ± 0.3 | 66.2 ± 0.2 | 66.9 ± 0.4 |
> | NMNC | 19.0 ± 1.6 | 62.7 ± 1.4 | 69.4 ± 0.7 | 70.4 ± 0.6 | 71.5 ± 0.4 |
>
> Following the reviewer’s suggestion, we also evaluated the fixed random subspace control on ImageNet. It performed substantially worse than both VNC and NMNC (best test accuracy: Random 25.7 ± 0.1, VNC 35.0 ± 0.2, NMNC 39.0 ± 0.4). We hypothesize that the larger gap with VNC (compared to CIFAR-10) arises because the solution space for ImageNet is more constrained, making restriction to an random subspace particularly limiting.
>
> > **Practical scalability and computational overhead of NMNC**
>
> We will clarify in the revised version that the current implementation is not scalable to arbitrary modern architectures. In our ImageNet implementation, the auxiliary GPU persistently stores about 1.5 GB of PCA bases and transiently another 1.5GB of accumulated activations for PCA updates with higher transient memory peaks during PCA computation (at least 6.5GB), while the training GPU stores a 1.5 GB mirrored copy of PCA bases. Despite this, the training time is largely amortized by running PCA asynchronously on the second GPU: backprop $154.81 \pm 0.16$ s, NMNC $153.52 \pm 2.08$ s, and VNC $235.48 \pm 0.46$ s per epoch (excluding the common data loading time). NMNC is substantially faster than VNC because perturbations are sampled in a low-dimensional manifold rather than the full activation space. Also, NMNC's forward pass runs under torch.no_grad() with gradients computed through the learned feedback path rather than a full autograd graph, which explains why it could be slightly faster than backprop.
>
> > **Performance dependence on the chosen manifold dimension d_l**
>
> We thank the reviewer for pointing this out. We studied the dependence of the task performance and gradient alignment on the manifold dimensionality in Fig. 9. As $d_l$ increases from small values, performance improves rapidly and then saturates; specifically, test accuracy plateaus once $d_l$ reaches roughly 512-512-512-128 for Conv1-3/FC1, which motivated our default choice. For the ImageNet experiment, the specific manifold dimensions were chosen because increasing the dimensionality further made PCA computation unable to keep up with training. We expect that using larger manifold dimensions would improve the performance of NMNC if PCA computation can keep up.
>
> > **Model-performance-controlled Brain Score evaluation**
>
> We thank the reviewer for mentioning this potential compound. To address it, we evaluated Brain Scores on the performance-matched NMNC models (checkpoints at epoch 11). As shown below, we found that they still significantly outperformed VNC. In addition, we also examined their Conv1 kernels and found that they do not exhibit the high-frequency patterns as VNC does. We quantified this as the average discrete Fourier transform for $f > 0.3$ ("DFT").
> | Scores ↓ \\ Models → | NMNC | Performance-matched NMNC | VNC |
> | --- | --- | --- | --- |
> | V4 | 0.519 ± 0.006 | 0.497 ± 0.002 | 0.475 ± 0.001 |
> | IT | 0.457 ± 0.003 | 0.438 ± 0.012 | 0.423 ± 0.006 |
> | Behavior | 0.169 ± 0.007 | 0.162 ± 0.006 | 0.127 ± 0.017 |
> | DFT (x $10^{-3}$) | 2.93 ± 0.24 | 2.95 ± 0.20 | 5.96 ± 0.04 |
> | Test Acc (%) | 39.0 ± 0.3 | 34.5 ± 0.4 | 35.0 ± 0.2 |
>
>
> > **Non-uniform algorithmic details of NMNC across experiments**
>
> We appreciate this observation and will clarify the framing. NMNC refers to a general principle (restricting perturbations to the neural manifold) that can be instantiated within different noise-correlation frameworks (e.g., direct feedback or layerwise Weight Mirror). We used direct feedback on CIFAR-10 for biological plausibility (it does not require reciprocal connections between layers), but realize that we could have used layerwise Weight Mirror also for CIFAR-10 to make the algorithmic details more uniform across scale.

---

> > ### Author Rebuttal · Reviewer_mbXq · 2026-04-05
> >
> > I acknowledge the authors' response to my queries. I am currently re-evaluating the manuscript based on their clarifications and will update my score to reflect their feedback.

---

### Official Review · Reviewer_iRav · 2026-03-13

**Soundness:** 4
**Presentation:** 4
**Significance:** 3
**Originality:** 3
**Overall Recommendation:** 5
**Confidence:** 4

**Summary:**

The paper proposes a well extended version of noise correlation methods for an important question in neuroscience and deep learning, based on the idea of constraining the gradient on neural manifold (NMNC), in opposition of isotropic perturbation in the full activation space

The simple but brilliant idea is the Jacobian of the network output with respect to  layer activations approximately lies in the neural manifold, so gradient estimation can be done in that lower dimensional subspace. The authors give a heuristic theoretical argument linking training dynmics to Jacobian–manifold alignment and derive a variance based analysis compring manifold restricted perturbations to isotropic noise.
Empirical validation has been provided on CIFAR-10 dataset, across ImageNet scales, and RNNs; showing better performance and sample efficiency compared to vanilla noise correlation.

The structure of paper is clear and easy to read. I find this work of interest to many colleagues in the field of computational neuroscience as well as deep learning. This paper is a clear example of meaningful NeuroAI contribution regardless of publication in this venue.

**Compliance With Llm Reviewing Policy:**

Affirmed.

**Key Questions For Authors:**

1) Can the authors provide a more formal characterization of when the Jacobian aligns with the neural manifold?

2) How sensitive is NMNC to errors in the manifold estimtion (for example when PCA does not capture the relevant subspace well)?

3) Does the advantage of NMNC still hold if the manifold dimension increases with network width or if the manifold assumption does not strictly hold?

4) How does NMNC compare with other varince reduction approaches for perturbation learning, such as structured noise or low rank perturbations?

5) What is the actual computational overhed of maintaining the manifold estimate in large models?

**Limitations:**

1) Although the theoretical sections are intuitive there is no theoretical analysis providing formal optimization guarantees. At least authors can clarify the if they solely rely on simulation evidences for optimization guarantee, or reframe the introduction claim regarding balanced between theory and experimental section.

2) The Jacobian–manifold alignment assumption is mainly empirical, and it is not clear how general this observation is across architectures. Maybe authors can better justify their assumption (either previous works or additional evidences )

3) Minor: The method relies on linear manifold approximation using PCA, which has been shown to not fully capture geometrical properties of actual data in comparison to intrinsic dimensionality methods and non-linear dimensionality reductions (Umap, TwoNN, etc).

**Strengths And Weaknesses:**

Strength:
Elegant conceptual idea: using low dimensional neural manifolds to improve perturbation based learning is intuitive and well motivated.

NMNC is simple and seems easy to implement on top of existing noise correlation methods.Relatively broad empirical evalution: experiments include CNN, ImageNe scale models and also RNNs. The paper gives a useful statstical interpretation of perturbation learning through covariance preconditioned gradients.


Weekness:
The main structural claim about the Jacobian aligning with the neural manifold is not really formally proven in the paper and it is  mainly  supported by empirical results and intuition rather than a rigorous argument.

Do we have guarantee that Jacobian always lie in a low dimensional space ? This is not very trivial concept to be assumed, it would be great if author either explain it or provide extended simulations.

---

> ### Author Rebuttal · Authors · 2026-03-29
>
> We thank the reviewer for the positive assessment and thoughtful questions. We answer them point-by-point below.
>
> > **Can the authors provide a more formal characterization of when the Jacobian aligns with the neural manifold?**
>
> We agree that our argument is heuristic rather than a formal guarantee, and will revise the paper to make that clearer. Because downstream weights and error signals reweight activation directions in a task-, architecture-, and history-dependent way, we do not claim exact containment of the Jacobian in the activity manifold. A more precise statement is that training induces preferential alignment of the Jacobian with the neural manifold relative to random subspaces, consistent with Eq. (2), Fig. 1, and footnote 2.
>
> To strengthen this empirically, we will add two results.
>
> First, following footnote 2, we repeated Fig. 1 using incremental PCA over activation history. This modestly increased Jacobian variance explained (epoch 100, 500 PCs: 0.58/0.53/0.33/0.88 → 0.61/0.53/0.35/0.92 for Conv1/2/3/FC1), indicating that current activations slightly understate alignment without changing the conclusion.
>
> Second, we repeated the analysis for ImageNet/AlexNet. Before training, variance explained was near random (0.01–0.06 for Conv1–5 with 2000 PCs; 0.24/0.24 for FC1/2 with 1000 PCs), but increased substantially after training began (epoch 1: 0.11/0.18/0.87/0.68/0.23 for Conv1–5; 0.58/0.36 for FC1/2). Thus, while we do not claim a theorem, results on both tested architectures support the narrower claim that training drives Jacobian alignment with the activity manifold.
>
> > **How sensitive is NMNC to errors in the manifold estimation (for example when PCA does not capture the relevant subspace well)?**
>
> NMNC should degrade gracefully with manifold-estimation error. The sensitivity is governed by how much of the true gradient lies inside the estimated subspace. In our analysis, the expected squared cosine with the true gradient scales approximately as $\alpha_l k /(k + d_l + 1)$ (Eqn. (12)), where $\alpha_l = \|P_l g_l\|^2/\|g_l\|^2$ is the fraction of gradient energy captured by the manifold projector (Eqn. (10)). Thus, if PCA returns an imperfect basis $\hat{P}_l$, the method degrades mainly through a smaller captured fraction $\hat{\alpha}_l = \|\hat{P}_l g_l\|^2/\|g_l\|^2$, leading to a graceful loss of benefit. Empirically, Fig. 9 is consistent with this picture: performance changes smoothly with manifold dimension rather than collapsing abruptly.
>
> > **Does the advantage of NMNC still hold if the manifold dimension increases with network width or if the manifold assumption does not strictly hold?**
>
> NMNC is advantageous whenever the estimated manifold captures a substantial fraction of gradient energy while remaining much lower-dimensional than the full activation space. In our theory this corresponds to large $\alpha_l$ and small $d_l/n_l$. Exact manifold containment is not required. However, if $d_l$ scales proportionally with $n_l$, then $d_l/n_l$ will no longer be small and the advantage should diminish.
>
> > **How does NMNC compare with other variance reduction approaches for perturbation learning, such as structured noise or low-rank perturbations?**
>
> NMNC is related in spirit to structured-noise and low-rank perturbation methods, but differs in a key respect: it restricts perturbations to an activity-defined, functionally relevant subspace rather than an arbitrary fixed low-rank subspace. This matters because low-rank perturbations help only insofar as the chosen subspace aligns with task-relevant gradient directions; otherwise they mainly improve variance prefactors or implementation cost, without reducing the effective dimensionality of the estimation problem (Appendix F). NMNC instead uses the network’s activity geometry to identify such an aligned subspace.
>
> > **What is the actual computational overhead of maintaining the manifold estimate in large models?**
>
> In our ImageNet implementation, the extra GPU for manifold estimation persistently stores 1.5 GB of PCA bases and transiently another 1.5 GB of accumulated activations for PCA updates, with higher transient peaks during PCA computation (at least 6.5 GB). The training GPU stores a 1.5 GB mirrored copy of the PCA bases. Despite this, the training time is largely amortized by running PCA asynchronously on the extra GPU: backprop $154.81 \pm 0.16$ s, NMNC $153.52 \pm 2.08$ s, and VNC $235.48 \pm 0.46$ s per epoch (excluding common data-loading time). NMNC is substantially faster than VNC because perturbations are sampled in a low-dimensional manifold rather than the full activation space.
>
> On the reviewer’s minor point, we agree that PCA is only a linear approximation to the manifold. We chose it as a simple online estimator to test the principle, and we view nonlinear or local manifold estimators as promising future directions.

---

> > ### Author Rebuttal · Reviewer_iRav · 2026-04-02
> >
> > Thanks the detailed and direct responses. Looking forward seeing the camera ready version (with the empirical evidences on alignment of Jacobin with neural manifold)
> >
> > I will increase the confidence score to 5.

---

> > > ### Author Response · Authors · 2026-04-06
> > >
> > > We thank the reviewer again for the careful reading of our manuscript and the constructive discussion. We will incorporate the additional empirical evidence on Jacobian–manifold alignment in the revised version as suggested.

---

### Decision · Program_Chairs · 2026-04-30

**Decision:**

Accept (regular)

**Comment:**

The paper proposes a neural manifold noise correlation (NMNC) method for training neural networks. The proposed approach relies on the fact that the Jacobian of the network output with respect to layer activations approximately lies in the neural manifold, so gradient estimation can be done in that lower-dimensional subspace. The reviewers highly appreciate this simple and yet elegant idea. Most reviewers note that the paper is well written, with sufficient theoretical justification for the approach and convincing numerical evidence for practical neural architectures. One reviewer was initially critical of the empirical studies, baselines compared with, and scalability concerns. The authors adequately address this comment in their rebuttal, which appears to have been accepted by the reviewer.